# Psychological distress among Japanese high school students during the COVID-19 pandemic: An energy landscape analysis

Daiki Tatematsu[1☯], Naotoshi Nakamura[1,2☯], Masato S. Abe[3,4,5☯], Tetsuo Ishikawa[6,7,8,9,10☯], Takahiro Ezaki[11☯], Lin Cai[12,13☯], Eiryo Kawakami[7,8,14,15], Kazuyuki Aihara[16], Atsushi Nishida[17], Naohiro Okada[16,18], Naoki Masuda[19], Kiyoto Kasai[16,18], Shinsuke Koike[13,16,20☯*], Shingo Iwami[1,15,16,21, 22,23,24☯*]

1 Interdisciplinary Biology Laboratory (iBLab), Division of Natural Science, Graduate School of Science, Nagoya University, Nagoya, Japan, 2 Department of Data Science, Yokohama City University, Kanagawa, Japan, 3 Faculty of Culture and Information Science, Doshisha University, Kyoto, Japan, 4 Center for Advanced Intelligence Project, RIKEN, Chuo-ku, Tokyo, Japan, 5 CBS-TOYOTA Collaboration Center, RIKEN, Wako, Saitama, Japan, 6 Division of Applied Mathematical Science, Center for Interdisciplinary Theoretical and Mathematical Sciences (iTHEMS), RIKEN, Yokohama, Japan, 7 Predictive Medicine Special Project (PMSP), RIKEN Center for Integrative Medical Sciences (IMS), RIKEN, Yokohama, Japan, 8 Department of Artificial Intelligence Medicine, Graduate School of Medicine, Chiba University, Chiba, Japan, 9 Department of Extended Intelligence for Medicine, The Ishii-Ishibashi Laboratory, Keio University School of Medicine, Tokyo, Japan, 10 Collective Intelligence Research Laboratory, Graduate School of Arts and Sciences, The University of Tokyo, Tokyo, Japan, 11 Research Center for Advanced Science and Technology, The University of Tokyo, Tokyo, Japan, 12 Division of Information Science, Nara Institute of Science and Technology, Nara, Japan, 13 Center for Evolutionary Cognitive Sciences, Graduate School of Arts and Sciences, The University of Tokyo, Tokyo, Japan, 14 Institute for Advanced Academic Research (IAAR), Chiba University, Chiba, Japan, 15 Interdisciplinary Theoretical and Mathematical Sciences Program (iTHEMS), RIKEN, Saitama, Japan, 16 The International Research Center for Neurointelligence (WPI-IRCN), Institutes for Advanced Study, University of Tokyo, Tokyo, Japan, 17 Unit for Mental Health Promotion, Research Center for Social Science and Medicine, Tokyo Metropolitan Institute of Medical Science, Tokyo, Japan, 18 Department of Neuropsychiatry, Graduate School of Medicine, University of Tokyo, Tokyo, Japan, 19 Gilbert S. Omenn Department of Computational Medicine and Bioinformatics and Department of Mathematics, University of Michigan, Ann Arbor, Michigan, United States of America, 20 Department of Child Neuropsychiatry, Graduate School of Medicine, The University of Tokyo, Tokyo, Japan, 21 Institute of Mathematics for Industry, Kyushu University, Fukuoka, Japan, 22 Institute for the Advanced Study of Human Biology (ASHBi), Kyoto University, Kyoto, Japan, 23 NEXT-Ganken Program, Japanese Foundation for Cancer Research (JFCR), Tokyo, Japan, 24 Science Groove Inc., Fukuoka, Japan

☯ These authors contributed equally to this work.
* c-koike@g.ecc.u-tokyo.ac.jp (SK); iwami.iblab@bio.nagoya-u.ac.jp (SI)

## Abstract

### Background

The stay-at-home orders, lockdowns, and states of emergency of the Coronavirus Infectious Disease emerged in 2019 (COVID-19) pandemic have affected the mental health of school-aged children. Previous reports of psychological distress in adolescents during the pandemic have been mixed, however, with some reports showing increases in psychological distress and others suggesting decreases. To accurately

**Data availability statement:** The data used in this study are part of the pn-TTC project and contain sensitive individual information collected from adolescents. For ethical and privacy reasons, the data cannot be made publicly available. Therefore, the data will be available after a review process by the pn-TTC projects. Please direct requests for data to Department of Neuropsychiatry, The University of Tokyo (https://value.umin.jp/data-resource.html). The programming code we used to calculate the energy landscape is a Python version of the MATLAB code available from https://github.com/tkEzaki/energy-landscape-analysis along with a tutorial. The Python code is available on GitHub (https://github.com/ttttmmttddiikk/Energy_Landscape_Analysis) and Zenodo (DOI: https://doi.org10.5281/zenodo.17585043).

**Funding:** This study was supported in part by Scientific Research (KAKENHI) B JP23H03497 (to S.I.) and JP24K02378 (to S.K.); Grant-in-Aid for Transformative Research Areas JP22H05215 (to S.I.), JP23H03877 (to S.K.), JP21H05171 and JP21H05174 (to K.K.); Grant-in-Aid for Challenging Research (Exploratory) JP22K19829 (to S.I.); AMED CREST JP19gm1310002 (to S.I.); AMED Research Program on Emerging and Re-emerging Infectious Diseases JP22fk0108509 (to S.I.), JP23fk0108684 (to S.I.), JP23fk0108685 (to S.I.); AMED Research Program on HIV/AIDS JP22fk0410052 (to S.I.); AMED Program for Basic and Clinical Research on Hepatitis JP22fk0210094 (to S.I.); AMED Program on the Innovative Development and the Application of New Drugs for Hepatitis B JP22fk0310504h0501 (to S.I.); AMED Strategic Research Program for Brain Sciences JP22wm0425011s0302 (to S.I.); AMED Brain/MINDS Beyond project JP18dm0307001 (to K.K.), JP18dm0307004 (to K.K. and S.K.), and JP18dm0307009 (to K.A.); AMED Brain/MINDS project JP19dm0207069 (to K.K. and S.K.); AMED Brain/MINDS 2.0 JP24wm0625001 (to K.K.) and JP24wm0625302 (to S.K. and S.I.); JST MIRAI JPMJMI22G1 (to S.I.); Moonshot R&D JPMJMS2021 (to K.A., S.I. and K.K.) and JPMJMS2025 (to S.I.); Institute of AI and Beyond at the University of Tokyo (to K.A.); Shin-Nihon of Advanced Medical Treatment Research (to S.I.); SECOM Science and Technology Foundation (to S.I.); The Japan

assess the impact of the pandemic, we need to be able to compare psychological assessments longitudinally, both before and during the pandemic. However, current statistical methods have limitations for reconstructing the complex trajectory of psychological states as captured by short-item questionnaires.

## Methods and findings

In this study, we analyzed monthly Kessler 6-item Psychological Distress Scale (K6) questionnaire responses collected from 16- to 18-year-old high school students participating in the population-neuroscience Tokyo TEEN Cohort (pn-TTC) in Japan (1,278 responses from 84 participants). Participants included 42 males and 42 females. The pn-TTC is a population-based longitudinal study conducted in Tokyo, Japan that follows children to investigate their developmental and mental health trajectories. In addition to conventional statistical approaches that summarize multiple questionnaire items into a composite score, we applied "energy landscape analysis," a method derived from statistical physics that models multivariate psychological states as a dynamic system of interactions among K6 questionnaire items, to visualize longitudinal changes in psychological distress before and during the COVID-19 pandemic (July 2019 to September 2021). Here, we define the depressive and healthy states as configurations in which all six K6 items are above or below each participant's individual mean, respectively. Before the pandemic, the healthy state occurred 11.0 times as frequently as the depressive state. In contrast, during the pandemic, the relative frequency of the healthy state increased to 18.2, 18.5, and 15.0 times that of the depressive state, respectively. The evolving energy landscape revealed an association between the pandemic period and a lower likelihood of being in a depressive state. We also identified two groups of students with different K6 dynamics and energy landscapes. The first group consisted of 61 participants whose total K6 score was relatively low (less than 5) and stable over time, and the second group consisted of 23 participants whose total K6 score was higher (with most being higher than 5) and less stable. The latter group showed a greater change in cortical thickness in the caudal part of the middle frontal gyrus (cMFG) ($t$-statistic $= -2.36$, $p$-value $= 0.019$, $q$-value $= 0.048$) and the temporal pole (TP) ($t = 3.08$, $p = 0.0023$, $q = 0.012$), as measured by magnetic resonance imaging, in the direction of accelerated adolescent brain development. Because all participants lived in Tokyo, generalizability remains limited, and as the association between psychological states and brain development is descriptive, future studies in diverse cohorts are needed to examine causality.

## Conclusions

By revealing associations between the COVID-19 pandemic and lower levels of psychological distress and healthier mental health states, our work demonstrates the potential of using dynamical systems theory, such as the energy landscape analysis,

Prize Foundation (to S.I.). URL of funder websites: KAKENHI https://www.jsps.go.jp/english/e-grants/ AMED https://www.amed.go.jp/en/index.html JST MIRAI https://www.jst.go.jp/mirai/en/index.html Moonshot R&D https://www.jst.go.jp/moonshot/en/ Institute of AI and Beyond at the University of Tokyo https://beyondai.jp/?lang=en Shin-Nihon of Advanced Medical Treatment Research http://www.shinnihon-zaidan.jp/ SECOM Science and Technology Foundation https://www.secomzaidan.jp/ The Japan Prize Foundation https://www.japanprize.jp/en/index.html. The funders had no role in study design, data collection and analysis, decision to publish, or preparation of the manuscript.

**Competing interests:** The authors have declared that no competing interests exist.

**Abbreviations:** B, beta; CI, confidence interval; cMFG, cortical thickness of the caudal part of the middle frontal gyrus; COVID-19, Coronavirus Infectious Disease emerged in 2019; CT, cortical thickness; df, degrees of freedom; Diff., difference; DOI, Digital Object Identifier; FDR, false discovery rate; GAMM, general additive mixed model; GHQ, General Health Questionnaire; GHQ-28, 28-item version of the GHQ; GLM, general linear model; GLMMs, general linear mixed models; G1, group 1; G2, group 2; HCP, Human Connectome Project; IQ, intelligence quotient; K6, Kessler 6-item Psychological Distress Scale; LCA, latent class analysis; MRI, magnetic resonance imaging; p, *p*-value; pn-TTC, population-neuroscience Tokyo TEEN Cohort; q, *q*-value (the false discovery rate-adjusted *p*-value); QC, quality control; SD, standard deviation; S.E., standard error; SES, socioeconomic status; SoE, state of emergency; SSE, sum of squared errors; t, *t*-statistic; TP, temporal pole; TS, traveling subject; TTC, Tokyo TEEN Cohort; T1w, T1-weighted; T2w, T2-weighted; WHO, World Health Organization.

to interpret health and disease metrics in psychology and psychiatry. This approach may improve mental health surveillance for the next pandemic.

## Author summary

### Why was this study done?

- The mental health of school-aged children has been affected by pandemic-related policies such as the stay-at-home orders, lockdowns, and states of emergency.

- Longitudinal comparisons of psychological assessments before and during the pandemic are essential to accurately evaluate its impact on adolescent mental health.

- Conventional statistical methods have limitations in capturing the complex dynamics of psychological states based on questionnaires, highlighting the need for new analytical approaches.

- Energy landscape analysis, a method derived from statistical physics, has the potential to capture the complex dynamics of psychological states.

### What did the researchers do and find?

- We applied the energy landscape analysis to monthly Kessler 6-item Psychological Distress Scale (K6) questionnaire responses collected from July 2019 to September 2021, spanning both the pre-pandemic and pandemic periods.

- Analysis revealed that during the COVID-19 pandemic, adolescents were less likely to experience a depressive state and more likely to transition to a healthier psychological state.

- We identified two groups of students with different K6 dynamics and energy landscapes. The group whose total K6 score was higher and less stable showed greater brain developmental change in cortical thickness in the caudal part of the middle frontal gyrus (cMFG) and temporal pole (TP).

### What do these findings mean?

- The COVID-19 pandemic was associated with a reduction in psychological distress and healthier mental health states among adolescents in this study.

- Dynamical systems approaches such as the energy landscape analysis may improve mental health monitoring during future pandemics.

- Main limitations include the fact that the participants in this cohort lived in Tokyo only, and the generalizability of our findings to adolescents living in other local areas and countries needs to be confirmed. Additionally, the association we found between psychological states and brain development remains descriptive, and further experimental or interventional studies are needed to validate causality.

## Introduction

The Coronavirus Infectious Disease emerged in 2019 (COVID-19) pandemic, a global emergency unprecedented in decades, has had a profound and widespread impact on people of all ages, including adults, adolescents, and children. Most notably, many countries implemented movement restrictions, such as stay-at-home orders and lockdowns, during the early stages of the pandemic. In Japan, states of emergency were declared that closed schools and other institutions. While these policies were generally effective in reducing the spread of COVID-19, they came at the cost of mental health outcomes [1,2]. To learn from the current pandemic and apply these lessons to the next one, we need to be able to assess the extent of psychological distress due to the pandemic and how mental health changed over time.

Recent studies have discussed the impact of social distancing due to the COVID-19 pandemic and infection on psychological outcomes such as depression and anxiety symptoms, psychological well-being, suicidal ideation and behavior, and loneliness [3–6]. Children and adolescents were particularly affected, with school closure significantly decreasing physical activity and social interaction [7]. Most reports showed increased psychological distress during the pandemic [8,9], but some conversely showed better mental health outcomes, suggesting relief from routine psychological stress experienced during school and social activities [10–12]. These findings suggest that adolescents may show heterogeneous patterns of response to a dramatic environmental change. However, because most of the surveys were conducted after the COVID-19 pandemic had already begun, we do not know how adolescents' psychological distress varied over time from the onset of the pandemic. Findings from cohort studies that began before the pandemic can be used to illustrate the trajectory of psychological distress. For example, one cohort study that assessed psychological distress in adults before and repeatedly during the pandemic showed four distinct trajectories of distress: continuously low, temporarily elevated, repeatedly elevated, and continuously elevated [13].

Self-report questionnaires are a common method for assessing the psychological status of human participants, and repeated measures can provide insight into the dynamic fluctuations in psychological distress during the COVID-19 pandemic. However, this method has limitations. Increasing the number of survey points may decrease the quality of the responses and result in a higher attrition rate. To address this limitation, shorter self-report questionnaires are needed, but this may decrease sensitivity and specificity. For example, the General Health Questionnaire (GHQ) [14] originally consisted of 60 items, and shorter versions are now available to increase response rates. The Kessler 6-item Psychological Distress Scale (K6) [15], which assesses psychological distress in six items only, is now commonly used; however, it assesses only one factor of psychological status and may have reduced sensitivity and specificity compared with the 28-item version of the GHQ (GHQ-28) [16], which assesses four factors (somatic symptoms, anxiety and insomnia, social dysfunction, and severe depression). These limitations are difficult to address simultaneously using current research methods in psychology; therefore, a new method is needed that can accurately reconstruct the complex trajectory of psychological distress from repeated responses to a short-item questionnaire.

In this study, we focus on high-school-aged adolescents and the impact of school closure [17] in the context of states of emergency, using the population-neuroscience Tokyo TEEN Cohort (pn-TTC) in Japan [18]. We analyze data collected through monthly mental health surveys of high school students conducted for 2 years before and during the pandemic. In addition to conventional statistical analysis, which aggregates multiple questionnaire items into a single score, we provide an in-depth analysis of the combination of each item using a method called "energy landscape analysis." The energy landscape analysis method allows us to intuitively visualize and interpret multidimensional categorical time-series data [19]. Rather than treating psychological distress as a static score that is simply "high" or "low," the energy landscape analysis captures the dynamic nature of mental states over time. This perspective focuses on the stability and variability of psychological states, providing an intuitive way to understand how mental health fluctuates and how individuals move between healthier and more distressed conditions. The analysis method explains the data by assigning an "energy" to each combination of variables (in this case, questionnaire items). Here, "energy" is a physical interpretation of probability, based on the Ising model in physics. We hypothesize that the energy landscape analysis can characterize how the pattern

and stability of adolescents' psychological states and mental health shifted before and during the pandemic, particularly in response to social restrictions such as school closure. Therefore, this study aims to examine whether the energy landscape analysis appropriately represents the underlying dynamics of adolescent mental health data and to explore how visualizing the distribution of psychological energy can provide insights into the impact of the pandemic on adolescent health. We also explore whether variations in the mental health survey and energy landscapes are linked to changes in adolescent brain development observed in brain magnetic resonance imaging (MRI) scans taken approximately 2 years apart.

## Methods

### Population-neuroscience Tokyo TEEN Cohort (pn-TTC)

**Study participants.** The pn-TTC [18] is an extension of the TTC study [20] and focuses on adolescent brain development and biomedical measurements. The TTC study is being conducted on 3,171 children randomly recruited from the resident registry in three municipalities of Tokyo, and the 479 pn-TTC participants were recruited from the TTC members (S1 Methods). Wave 1 (age 11 years) of the pn-TTC recruitment began in September 2013 and ended in February 2016. For the following Waves 2–4 (ages 13, 15, and 18, respectively), participant recruitment was conducted biennially. A total of 1,271 scans from 479 participants were performed between October 2013 and March 2023, and 1,211 scans from 471 participants were used after quality controls were implemented during image processing (S1 Fig)

In pn-TTC Wave 3 (conducted from May 2018 to September 2020), all participants were recruited for the monthly web-based questionnaire survey using their smartphones. A total of 109 participants provided written informed consent, and 1,278 responses from 84 participants who provided valid responses were used for the energy landscape analysis.

**Monthly web-based questionnaire survey (including K6).** The monthly web-based survey included the Japanese version of the K6 scale [21,22] and background questions. The K6 is a widely used scale of subjective psychological distress over the past 30 days. Each response is scored from 0 (never) to 4 (always). In common use, the total score of 6 items (range, 0–24) is considered. For this study, we used the background questions on the number of school days per week. Each participant was asked to select one of the following options for school days in the past 30 days: not commuting to school, 1 day per week, 2 days per week, 3 days per week, 4 days per week, 5 days per week, 6 days per week, or 7 days per week. In accordance with the School Education Act Enforcement Order in Japan [23], summer vacation was assumed to be in August and winter vacation was assumed to be in January.

### Image acquisition and processing

For the four waves of data collection (from September 2013 to June 2023), two 3-T scanners and three acquisition procedures were used (S1 Fig). Only T1-weighted images were acquired for procedures 1 and 2, whereas T1-weighted and T2-weighted images were acquired for procedure 3 (S2 Methods and S1 Table). When only T1-weighted images were available, the legacy style of the Human Connectome Project pipeline was used for image preprocessing. In cases in which both T1-weighted and T2-weighted images were available, the standard style of the Human Connectome Project pipeline was used for image preprocessing [24]. Then, we extracted the 75 features of cortical thickness, cortical surface area, and subcortical volume using the Desikan–Killiany atlas. Traveling subject harmonization was performed to diminish the procedural difference for each brain feature (S2 Methods) [10,25,26]. Quality controls were performed in each preprocessing step in a standardized manner (S2 Methods) [10,25].

### The first COVID-19 case, the declarations of a state of emergency, and the definition of the four periods

Wuhan Municipal Health Commission, China, reported a cluster of pneumonia cases in Wuhan, Hubei Province, China, to the World Health Organization (WHO) on December 31, 2019 [27]. The first COVID-19 case in Japan was reported to WHO on January 16, 2020 [28]. The first declaration of a state of emergency in Tokyo, Japan, was from April 7, 2020, to May 25,

2020 [29]. The second was from January 8, 2021, to March 21, 2021; the third was from April 25, 2021, to June 20, 2021; and the fourth was from July 12, 2021, to September 30, 2021.

In this study, the four periods were defined as follows: (1) Period 1: July 2019 to December 2019. This is the period before the COVID-19 pandemic. (2) Period 2: January 2020 to May 2020. This period includes the first COVID-19 case in Japan and the first declaration of a state of emergency. (3) Period 3: June 2020 to December 2020. This is the period during which no state of emergency was declared. (4) Period 4: January 2021 to September 2021. This period includes the second, third, and fourth declarations of a state of emergency.

### Preprocessing of time-series K6 scores

First, the acquisition time of the K6 score data was rounded to the nearest day. Second, because the questionnaire asked about participants' status in the past 30 days, we extended the same data to the previous 30 days, assuming that the same status continued during this period. If the extension resulted in more than one data point on the same day, the data were averaged. Third, using gaussian process regression, the data were further interpolated to monthly data. The kernel function was defined using a combination of the sklearn.gaussian_process.kernels class in the python scikit-learn (https://scikit-learn.org), RBF() + ConstantKernel() + WhiteKernel(). Fourth, the data were binarized to perform the energy landscape analysis. The average was computed for each item per participant over the entire period. Using this average, the time series was binarized and assigned a value of 1 if above average and 0 otherwise. Values were set to 0 for items for which all values were equal. We express an individual response as an $N$-dimensional vector (hereafter referred to as a "response vector") whose components consist of binarized responses to $N$ questionnaire items. This preprocessing workflow is illustrated in S2 Fig.

### Stratifying participants using time-series K6 scores

Participants were stratified using $k$-means clustering based on preprocessed time-series K6 scores (Python scikit-learn). We also computed the within-cluster sum of squared errors (SSE) to assess the accuracy of the clustering. To see the difference in depressive symptoms at the times of the MRI scans at Waves 3 and 4 between the clustering groups, we used a logistic regression model (S2).

### Energy landscape analysis

The programming code we used to calculate the energy landscape is a Python version of the MATLAB code available from [30] along with a tutorial. The Python code is available on GitHub (https://github.com/ttttmmttddiikk/Energy_Landscape_Analysis) and Zenodo (https://doi.org/10.5281/zenodo.17585043).

**Pairwise maximum entropy model.** We fit the pairwise maximum entropy model to the pooled individual response data in the same manner as in the previous studies [31]. Let $\mathcal{S}$ be the set of all the possible "states" of $N$-dimensional response vectors, where $N$ is the number of questionnaire items ($N = 6$ in the case of K6). $\mathcal{S}$ has $2^N$ elements in total. For all $\boldsymbol{\sigma} = (\sigma_1, \ldots, \sigma_N) \in \mathcal{S}$, we assign the probability $P_{\text{model}}(\boldsymbol{\sigma})$ such that

$$\sum_{\boldsymbol{\sigma} \in \mathcal{S}} P_{\text{model}}(\boldsymbol{\sigma}) = 1$$

(1)

The Shannon entropy is given by

$$S(P_{\text{model}}) = -\sum_{\boldsymbol{\sigma} \in \mathcal{S}} P_{\text{model}}(\boldsymbol{\sigma}) \log P_{\text{model}}(\boldsymbol{\sigma}),$$

(2)

where log is a natural logarithm. We maximize the entropy (2) under equation (1) and the following constraints (3) and (4):

$$\langle \sigma_i \rangle_{\text{model}} = \langle \sigma_i \rangle_{\text{data}} \quad \text{for all } i \text{ with } 1 \leq i \leq N,$$

(3)

$$\langle \sigma_i \sigma_j \rangle_{\text{model}} = \langle \sigma_i \sigma_j \rangle_{\text{data}} \quad \text{for all } i, j \text{ with } 1 \leq i < j \leq N, \tag{4}$$

where $\langle \sigma_i \rangle_{\text{model}} = \sum_{\boldsymbol{\sigma} \in \mathcal{S}} \sigma_i P_{\text{model}}(\boldsymbol{\sigma})$, $\langle \sigma_i \rangle_{\text{data}} = \frac{1}{T} \sum_{t=1}^{T} \sigma_i^t$, $\langle \sigma_i \sigma_j \rangle_{\text{model}} = \sum_{\boldsymbol{\sigma} \in \mathcal{S}} \sigma_i \sigma_j P_{\text{model}}(\boldsymbol{\sigma})$, $\langle \sigma_i \sigma_j \rangle_{\text{data}} = \frac{1}{T} \sum_{t=1}^{T} \sigma_i^t \sigma_j^t$. Here, $\langle \ \rangle$ denotes the mean, and $\sigma_i^t \in \{0, 1\}$ denotes the $i$th component ($1 \leq i \leq N$) of the $t$th response vector ($1 \leq t \leq T$) in the pooled response data consisting of $T$ individual responses. Using the method of Lagrange multipliers, we obtain the following equation about the model probability:

$$P_{\text{model}}(\boldsymbol{\sigma}|\boldsymbol{h}, \boldsymbol{J}) = \frac{\exp[-E_{\text{model}}(\boldsymbol{\sigma}|\boldsymbol{h}, \boldsymbol{J})]}{\sum_{\boldsymbol{\sigma}' \in \mathcal{S}} \exp[-E_{\text{model}}(\boldsymbol{\sigma}'|\boldsymbol{h}, \boldsymbol{J})]}, \tag{5}$$

where

$$E_{\text{model}}(\boldsymbol{\sigma}|\boldsymbol{h}, \boldsymbol{J}) = -\sum_{i=1}^{N} h_i \sigma_i - \frac{1}{2} \sum_{i=1}^{N} \sum_{\substack{j=1 \\ j \neq i}}^{N} J_{ij} \sigma_i \sigma_j \tag{6}$$

is the energy, and $\boldsymbol{h} = (h_1, \ ..., \ h_N)$ and $\boldsymbol{J} = (J_{ij})$ $(i, j = 1, \ldots, N)$ are the parameters of the model ($\boldsymbol{J}$ is symmetric: $J_{ij} = J_{ji}$). Here, interactions higher than second order are ignored, hence the name pairwise maximum entropy model. Equations (5) and (6) match those of the Boltzmann distribution and the Ising model, respectively. The maximum-likelihood solution is expressed as

$$(\boldsymbol{h}^*, \boldsymbol{J}^*) = \arg \max_{\boldsymbol{h}, \boldsymbol{J}} \mathcal{L}(\boldsymbol{h}, \boldsymbol{J}),$$

where $\mathcal{L}(\boldsymbol{h}, \boldsymbol{J})$ is the likelihood given by

$$\mathcal{L}(\boldsymbol{h}, \boldsymbol{J}) = \prod_{t=1}^{T} P_{\text{model}}(\boldsymbol{\sigma}^t|\boldsymbol{h}, \boldsymbol{J}),$$

and $\boldsymbol{\sigma}^t = (\sigma_1^t, \ \sigma_2^t, \ldots, \sigma_N^t)$ is the $t$th response vector ($1 \leq t \leq T$) in the pooled response data. We maximize the likelihood by a gradient ascent scheme given by

$$h_i^{\text{new}} = h_i^{\text{old}} + \alpha \frac{\partial}{\partial h_i} \log \mathcal{L}(\boldsymbol{h}, \boldsymbol{J}) = h_i^{\text{old}} + \alpha T(\langle \sigma_i \rangle_{\text{data}} - \langle \sigma_i \rangle_{\text{model}})$$

and

$$J_{ij}^{\text{new}} = J_{ij}^{\text{old}} + \alpha \frac{\partial}{\partial J_{ij}} \log \mathcal{L}(\boldsymbol{h}, \boldsymbol{J}) = J_{ij}^{\text{old}} + \alpha T \left( \langle \sigma_i \sigma_j \rangle_{\text{data}} - \langle \sigma_i \sigma_j \rangle_{\text{model}} \right),$$

where the superscripts *new* and *old* represent the values after and before a single updating step, respectively, and $\alpha$ ($> 0$) is the learning rate. Updating was stopped when

$$\frac{\sqrt{\| \left( \frac{\partial}{\partial h_1} \log \mathcal{L}(\boldsymbol{h}, \boldsymbol{J}), \ \cdots, \ \frac{\partial}{\partial h_N} \log \mathcal{L}(\boldsymbol{h}, \boldsymbol{J}) \right) \|_2^2 \ + \ \| \left( \frac{\partial}{\partial J_{ij}} \log \mathcal{L}(\boldsymbol{h}, \boldsymbol{J}) \right) \|_F^2}}{N + N^2}$$

became less than the permissible error $\varepsilon$ (the first term in the numerator is the square of the 2-norm; the second term is the square of the Frobenius norm). We set $\alpha = 0.001$ and $\varepsilon = 0.005$. For the monthly data, we set $\varepsilon = 0.05$ to reduce computation time. Initial values of $h_1, \ ..., \ h_N$ and $J_{ij}$ $(i, j = 1, \ldots, N)$ were set to 0.

The model variables and parameters $\sigma$, $P_{model}$, $E_{model}$, $h$ and $J$ are particularly important throughout the study. They are listed with explanations in S3 Table.

**Accuracy indices.** To quantify the accuracy of the pairwise maximum entropy model fitted to the data, we used the following two accuracy indices [19,32,33]: The first accuracy index is defined by

$$\frac{I_2}{I_N} = \frac{S_1 - S_2}{S_1 - S_N},$$

where $S_k = -\sum_{\sigma \in S} P_k(\sigma) \log P_k(\sigma)$ is the Shannon entropy of the maximum entropy model incorporating correlations up to the $k$th order. The probability distribution $P_1(\sigma)$ represents the independent maximum entropy model in which we ignore any interactions between items by forcing $J_{ij} = 0$ for all $i, j = 1, \ldots, N$. Distribution $P_2(\sigma)$ represents the pairwise maximum entropy model. Distribution $P_N(\sigma)$ is equivalent to the distribution of the given data, $P_{data}(\sigma)$. The second accuracy index is defined by

$$r = \frac{D_{KL}(P_1(\sigma) \| P_N(\sigma)) - D_{KL}(P_2(\sigma) \| P_N(\sigma))}{D_{KL}(P_1(\sigma) \| P_N(\sigma))},$$

where $D_{KL}(P \| Q)$ is the Kullback–Leibler divergence between two distributions $P$ and $Q$.

**Assigning an energy to each state.** We calculated the energies for all $2^N$ states. For two states $\sigma_i$ and $\sigma_j$, if $E(\sigma_i) > E(\sigma_j)$, this means that $P(\sigma_i) < P(\sigma_j)$, and under the condition of detailed balance, a state transition is more likely to occur from $\sigma_i$ to $\sigma_j$ than from $\sigma_j$ to $\sigma_i$.

**Stable states.** A stable state is a state ($N$-dimensional vector) whose energy is lower than that of all $N$ adjacent states (i.e., states that differ by 1 bit) and is a local minimum. For all $2^N$ states, we examined whether each state was a stable state by comparing the energies of all $N$ adjacent states.

**Basin and basin size.** For all $2^N$ states except stable states, we determined which stable state was reached by repeatedly moving to the state with the lowest energy among adjacent states. Thus, each state belonged to the basin for the reached stable state. We defined the basin size as the number of states belonging to the basin.

**Disconnectivity graph.** A disconnectivity graph is a visualization method based on a dendrogram to represent the (dis)connectivity between stable states. In the disconnectivity graph, stable states are plotted as leaves and are located at a height on the $y$-axis corresponding to their energy values. The branching structure of the disconnectivity graph represents the energy barriers between any pair of stable states. Short branches were removed following the protocol shown in [34].

**Probabilities of being in a basin and of transitioning between basins.** Using the basins from the network, we determined which basins the data belonged to. The probability of being in a basin is calculated as the frequency with which the data fall into that basin. The transition probability from basin A to basin B is the conditional probability that if an individual is in basin A in a given month, the same individual will be in basin B the following month. It is calculated as [frequency of individuals who are in basin A in month $n$ and in basin B the following month] divided by [frequency of individuals in basin A in month $n$].

**Numerical simulations.** We performed numerical simulations to generate hypothetical transition sequences constrained by the obtained landscape [35]. We employed Gibbs sampling as follows. First, we set one of the stable states (000000 or 111111) as the initial state. Then, at each time step, a transition is attempted from the current state $\sigma_1$ to one of its adjacent states $\sigma_2$, selected with probability 1/6. The transition to the selected state occurs with probability $e^{-E_{model}(\sigma_2)}/(e^{-E_{model}(\sigma_1)} + e^{-E_{model}(\sigma_2)})$.

## Statistical analysis of brain image features

The analysis of adolescent brain trajectory and its relationship with the clustering was performed using R software version 4.3.2. To explore the adolescent developmental trajectory of each brain feature, we built a GAMM implemented with the

mgcv package (version 1.9-1) [36], including the main effect of sex as a linear term, the main effect of age as a smooth term using a penalized cubic regression spline and a basis function of 4, a tensor interaction term between age and sex, intracranial volume as a linear covariate, and a random intercept per participant (S2 Methods). Because the use of a subset of brain images (266 scans) from the 84 participants included in the energy landscape analyses may cause sampling bias in relation to the whole cohort (1,211 scans from 471 participants), we extracted the deviation from the nonlinear model for each scan and feature [37]. We then tested the difference between the groups from the clustering using GLMMs implemented with the lme4 package (version 1.1-35.1), including the main effect of group (G1 or G2), age × group interaction, child IQ, socioeconomic status, and handedness as covariates, and a random intercept per participant. Because we conducted 75 repeated tests, FDR correction was applied. To see whether the GHQ-28 scores at Wave 3 and/or the difference between waves was also associated with adolescent brain features, we further tested the same GLMMs. For the features with significant associations, we tested whether the group (G1 or G2) and/or the GHQ-28 scores could explain the brain difference and trajectory by performing the GLMMs adding the main effects of both variables and the interactions by age.

### Ethics statement

This study was approved by the ethics committee of the Faculty of Medicine, The University of Tokyo (No. 10069), and the research ethics committee of The University of Tokyo (No. 22-396). Written informed consent was obtained from participants in the monthly web-based survey. After written informed consent was obtained, the participants' parents were informed of their participation and had the opportunity to withdraw consent. For brain MRI scans, written informed consent was obtained from participants' primary caregivers and participants 15 years of age or older prior to participation in this study.

### STROBE guideline and TRIPOD statement

This study is reported as per the Strengthening the Reporting of Observational Studies in Epidemiology (STROBE) guideline (S1 Checklist) and the Transparent Reporting of a Multivariable Prediction Model for Individual Prognosis Or Diagnosis (TRIPOD) statement (S2 Checklist).

## Results

### Collection of K6 questionnaire responses from adolescents before and during the COVID-19 pandemic

To analyze changes in adolescents' emotional states during the COVID-19 pandemic, we obtained monthly responses to the K6 questionnaire [15], which measures nonspecific psychological distress, completed by 16- to 18-year-old high school students in Tokyo who participated in the pn-TTC from July 2019 to October 2021, starting 8 months before the first state of emergency declaration (Figs 1A and S1). Participants included 42 males and 42 females. The questionnaire contains 6 questions about the participants' psychological distress over the past 30 days, and each item is rated on a 5-point scale (Fig 1B). The results showed Pearson correlation coefficients ranging from 0.53 to 0.81 between items (Fig 1C), suggesting that the items of the K6 questionnaire, whose total score indicates the level of psychological distress, were moderately to strongly correlated, as shown in previous validation studies [38–40].

Conventional statistical analyses of K6 responses have mainly focused on the total score and have not been able to fully exploit the multidimensional structure and dynamic features of the questionnaire. In order to analyze the dynamics of K6 responses in an integrated manner, encompassing six dimensions, we employed a method referred to as an "energy landscape analysis". In this analysis, each questionnaire item is represented by a spin that takes the value of either 0 or 1 (in statistical physics, a spin usually takes the value of +1 or −1, but the mathematical model is the same except for differences in constants [41]). We assign an "energy" to each spin ($h_i$) and to the interaction of two spins ($J_{ij}$), and the total energy represents the probability of a specific combination of K6 responses (lower energy means higher probability; see Methods and S1 Note (Glossary) for further details).

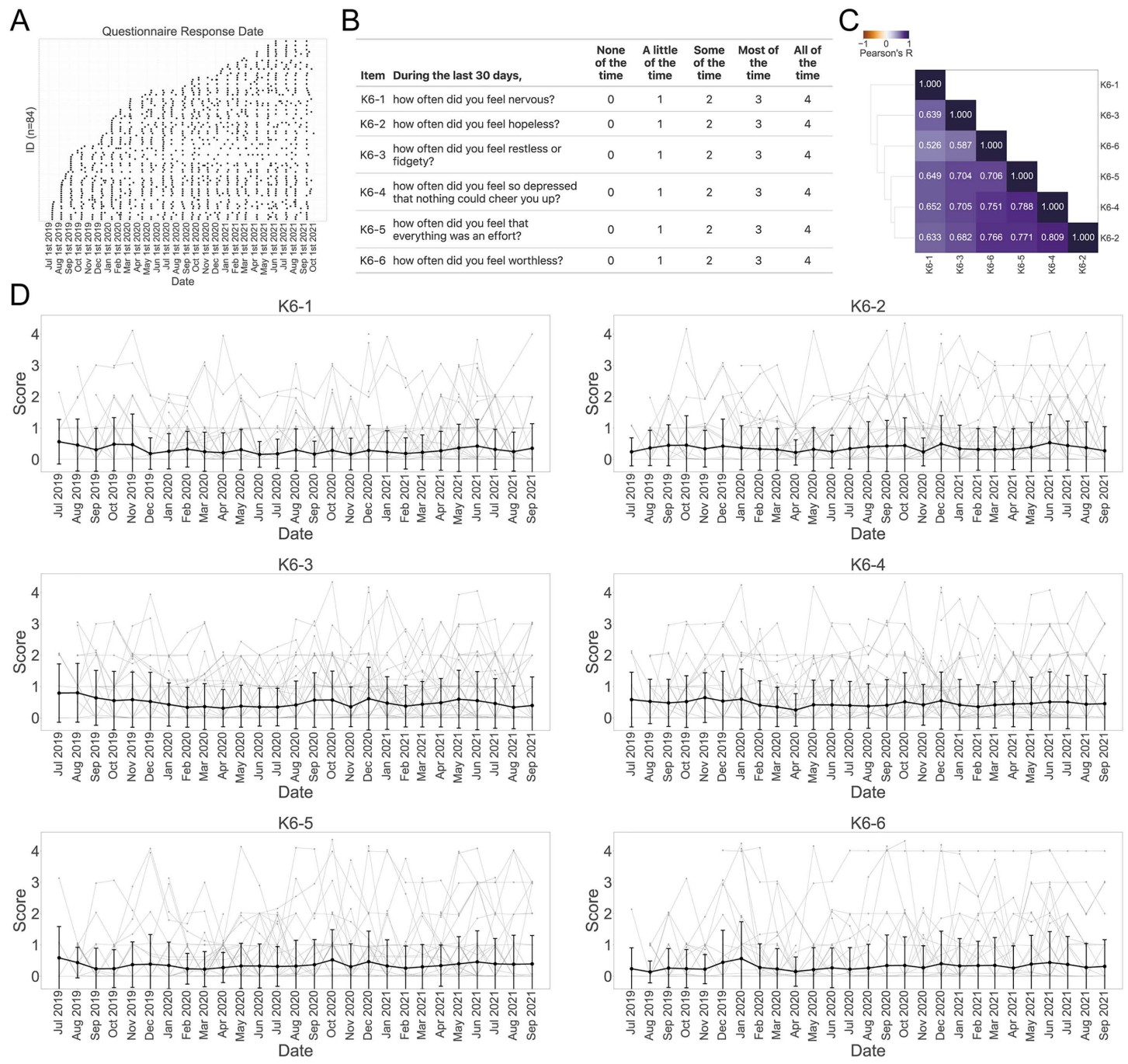

**Fig 1. Collecting K6 scores from high school students in Japan.** (A) The timeline of K6 responses is described for each participant ($n = 84$ participants and 1,278 total responses). Black dots indicate the date on which the response was received. **(B)** Questionnaire items and scales. **(C)** The correlation matrix between the item-level scores. Pearson's correlation coefficients are shown in pseudo-colors ranging from orange to purple. Each item is arranged based on hierarchical clustering. **(D)** The time series of the item-level scores after data cleaning (see Methods, S2 and S3 Figs). All items from K6-1 to K6-6 are plotted with time on the *x*-axis and score on the *y*-axis for each participant. The thick line represents the mean. Error bars indicate standard deviation (SD).

Thus, the collected and preprocessed questionnaire responses (Figs 1D, S2, and S4) were binarized according to the procedures described in the Methods (illustrated in S3A Fig), and the binarized K6 scores were used in subsequent analyses. Here, binarization cutoffs were set to the individual mean and were therefore allowed to vary between individuals and between questionnaire items. See S2 Note and S5 Fig for a comparison of different binarization methods.

## Energy landscapes for K6 scores during the COVID-19 pandemic

We first pooled the binarized K6 scores of all participants and constructed the energy landscape for the entire period, which we visualized as a "disconnectivity graph" (Methods and S3B Fig). We provide an explanation of a disconnectivity graph in Figs 2A and S1 Note. The accuracy of the model fitting is evaluated in S6A Fig.

The disconnectivity graph was drawn with the stable states shown in Fig 2B. Here, each state is represented by a six-bit number, each bit of which represents K6-1, K6-2, K6-3, K6-4, K6-5, and K6-6 from left to right. The stable states, defined as local minima in the energy landscape (i.e., state(s) with lower energy than all adjacent states), were 000000 (i.e., binarized responses to all 6 items were 0) and 111111 (i.e., binarized responses to all 6 items were 1), corresponding to the "healthy" and the "depressive" states (states of psychological distress). The fact that the two stable states had the same digits across all items is consistent with the relatively strong correlation of the K6 questionnaire items. Indeed, all the estimated coefficients of interaction ($J_{ij}$) were positive (S4 Table), which means that states become more stable when different digits take the same values. Furthermore, to quantify the uncertainty of the estimated parameters, we computed confidence intervals for both $h_i$ and $J_{ij}$ and confirmed that these intervals were notably narrow, indicating stable estimates (S7 Fig). We also confirmed stable parameter estimation based on the convergence of estimation errors and the change in parameter estimates across iterations (S8 Fig). The energy difference between 000000 and 111111 was $0 - 2.75 = -2.75$, meaning that the probability of 000000 was $e^{2.75} \approx 15.6$ times greater than that of 111111 (i.e., lower energy means higher probability).

To see how all states are related in the energy landscape, we also plotted a forest structure (called the graph of basins of attraction) in which each state is connected to an adjacent state with the lowest energy (Figs 2C and S3B). Here, each state is represented by a six-bit number; an adjacent state of state A differs from A by only one bit. The structure consisted of two disjoint trees (called basins of attraction, hereafter "basins") with root 000000 or 111111. Each basin consists of all states that lead to its root (stable state) by the steepest descent method, i.e., by moving downhill in the energy landscape. In other words, a basin is a set of states that are expected to be attracted to the same stable state. Therefore, the 000000 basin can be interpreted as the relatively healthier states (attracted to the healthy stable state 000000), and the 111111 basin as the relatively more depressive states (attracted to the depressive stable state 111111). The basin sizes of the 000000 and the 111111 basins were 40 and 24, respectively.

To explore the connection between students' depressive states and their social context, we examined differences across the four periods outlined by the declarations of states of emergency (Fig 2D), considering that social interactions at high schools primarily occur on school days. The first state of emergency was declared in Period 2 (January to May 2020), whereas the second, third, and fourth declarations took place in Period 4 (January to September 2021). Notably, the number of school days during the states of emergency, especially in the first one, was significantly small. When we compared the mean of the total K6 score across the four periods, there was a significant trend indicating a decrease in the K6 score in Period 2 and then a slight increase in Periods 3 and 4 (Jonckheere–Terpstra trend test, $p = 0.038$) (Fig 2E; see also S9 Fig). We speculate that an increase in time spent at home during the pandemic, especially in the first state of emergency, led to a reduction in social interactions. This reduction in social interactions may have helped to alleviate psychological distress for high school students, as previously mentioned in [42]. In addition, as demonstrated by our different questionnaire data presented in S10 Fig, we revealed a correlation between fewer school days per week and lower K6 scores.

However, the total K6 score alone does not reveal which combination of items contributed to the healthy and depressive states, nor how these states changed over time. To see this, we looked at individual K6 items and performed an

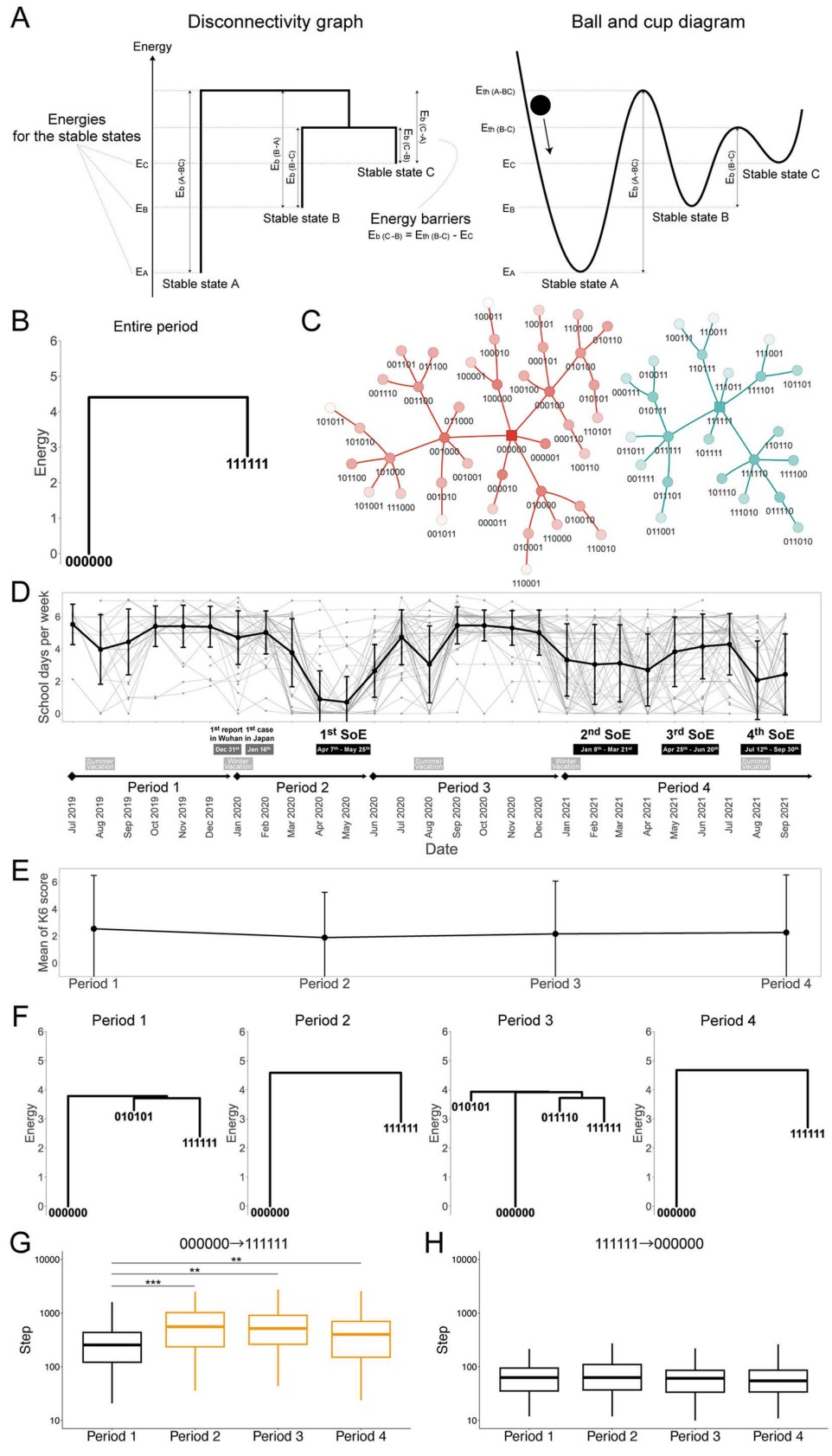

**Fig 2. Energy landscape analysis of time-series K6 questionnaire responses for all participants. (A)** An example of a disconnectivity graph (left) and the equivalent ball-and-cup diagram (right) are shown with explanations. Both display stable states and the energy barrier between them. Additional details on disconnectivity graphs can be found in S1 Note (Glossary). **(B)** The disconnectivity graph of the energy landscape for the entire period is shown. Each stable state is represented by a six-bit number consisting of the binarized values of K6-1, K6-2, K6-3, K6-4, K6-5, and K6-6 from left to right. The cutoff value of binarization was set to the individual mean; thus, 0 is below or equal to the individual mean and 1 is above the individual mean. **(C)** The graph of the basin of attraction obtained from the energy landscape analysis for the entire period. The stable states are plotted as squares. Each state is connected to the direction of attraction by an edge and each basin of attraction is shown in a different color. The basin 000000 is shown in red; the basin 111111 is shown in green. The density of the color corresponds to the magnitude of the energy. The darker the color, the lower the energy, and the lighter the color, the higher the energy. The six-bit number under each state shows the binarized values of K6-1, K6-2, K6-3, K6-4, K6-5, and K6-6 from left to right. **(D)** The definition of the four periods utilized in the analysis is explained, along with the number of school days per week for each participant. The thick line represents the mean. Error bars indicate standard deviation (SD). SoE refers to a state of emergency. **(E)** The mean of the total K6 scores for the 4 periods is shown. Error bars indicate standard deviation (SD). **(F)** The disconnectivity graphs of the energy landscape for the 4 periods are shown. **(G, H)** The simulated times from 000000 to 111111 or from 111111 to 000000 are compared between the 4 periods. 000000 (binarized responses of 0 to all six items) can be considered "healthy," whereas 111111 (binarized responses of 1 to all six items) represents the "depressive" state. Box plots show the median (center line), interquartile range (box), and range (whiskers). The upper outliers are hidden to improve visibility. Yellow indicates increase compared with Period 1.

energy landscape analysis on the pooled data for each of the 4 periods; the disconnectivity graphs were drawn similarly (Fig 2F; the accuracy is evaluated in S6B Fig). The stable states included 000000 and 111111 for all periods. There were also other stable states with higher energy (lower probability), with an energy difference of >3 from 000000 (i.e., more than 20 times lower probability). The energy of 000000 is 0 from the definition of the model. The energy of 111111 was lower in Period 1 (2.40) than in the other periods (2.90, 2.92, and 2.71), meaning the probability of 111111 was higher before the pandemic and became lower after the onset of the pandemic. Note that, when the K6 scores were analyzed item by item, use of traditional statistical methods such as the Jonckheere–Terpstra trend test, Tukey's test, or Mood's median test did not reveal significant differences among the different periods (S5 Table), suggesting that these methods were not sensitive enough to detect item-level differences. To further validate the findings from the energy landscape analysis, we performed a latent class analysis based on total K6 scores, which independently reproduced the same temporal pattern (see S3 Note and S11 Fig).

To examine how participants moved between different states over time, we simulated state transitions constrained by the energy landscape using Gibbs sampling (Methods), based on the simplifying assumption of detailed balance [43]. The logarithm of the average dwell time of each state was a linear function of its energy with a negative slope, consistent with the theoretical prediction (the probability of states follows a Boltzmann distribution in the model) (S12 Fig). Notably, when we started the simulation at 000000 and continued until the state reached 111111, the mean time from 000000 to 111111 became significantly longer in Periods 2–4 than in Period 1 (Fig 2G). In contrast, when we started the simulation at 111111 and continued until the state reached 000000, the mean time from 111111 to 000000 was not significantly different among the four periods (Fig 2H). These results suggest that it became more difficult to move from 000000 to 111111 after the pandemic started.

The above trend in the simulations was confirmed by a statistical analysis of the two basins (000000 and 111111) (S4 Note and S13AB Fig). The probability of staying in the healthy 000000 basin increased in Periods 2–4 compared to Period 1 (S13A Fig), and the monthly transition probability from the depressive 111111 basin to the healthy 000000 basin increased in Periods 2–4 compared to Period 1 (S13B Fig), suggesting that students were more likely to be in the healthy 000000 basin after the pandemic.

We further constructed energy landscapes on the pooled data for each month (from July 2019 to September 2021, S14A Fig), although we note that the accuracy of landscape construction may have been lower due to a small sample size (see also S15 Fig for confidence intervals of energy estimates). Nevertheless, the energy difference between 000000 (healthy state) and 111111 (depressive state) was negatively correlated with the mean number of school days per week ($r = -0.35$, $p = 0.071$) (S14BC Fig), again suggesting that students were less depressed when school days per week were fewer.

## Energy landscapes for diverse psychological distress to the COVID-19 pandemic

The patterns of psychological responses to crises, including to the COVID-19 pandemic, vary greatly among individuals [44]. To gain deeper insights into such diverse responses, we next stratified participants using *k*-means clustering, with a focus on the time series of total K6 scores (Figs 3A and S16). We identified two distinct groups: G1 consisted of 61 participants whose total K6 score was relatively low (less than 5) and stable over time, and G2 consisted of 23 participants whose total K6 score was higher (with most being higher than 5) and less stable (Figs 3B and S17) (Welch *t* test with Benjamini–Hochberg correction; $p = 2.9 \times 10^{-8}$, $3.8 \times 10^{-14}$, $5.8 \times 10^{-19}$, $1.1 \times 10^{-25}$ for Periods 1–4, respectively). Some participants in G2 maintained a total K6 score above 5 across all periods, while others had a total K6 score below 5 in at least one period (S9 Fig). The number of participants in the four periods, categorized by the clinically relevant cutoffs (5, 8, 13) of the total K6 score (the cutoff values were chosen based on [22]), is shown in S6 Table. G1 included 36 males and 25 females, while G2 included 6 males and 17 females. Similar to the findings for the K6, G2 showed higher scores on the 28-item version of the GHQ (GHQ-28) [16] (S1 Methods and S7 Table), which assesses more subjective affective symptoms than does the K6. G2 also showed a greater increase than G1 in the GHQ-28 score between the two brain MRI scans. No significant differences were observed between G1 and G2 in IQ or parental socioeconomic status (S7 Table).

We constructed separate energy landscapes for G1 and G2 and compared the basins derived from the energy landscapes (Fig 3C and 3D). The stable states for both groups were 000000 (i.e., healthy state) and 111110/111111 (i.e., depressive(-like) state). The sizes for the 000000 and 111110/111111 basins were 47 and 17 for G1, and 33 and 31 for G2, respectively.

We compared the disconnectivity graphs of G1 and G2 drawn from the energy landscape (Fig 3E and 3F: the accuracy indices are shown in S6C–S6E). The energy of 111111 or 111110 (depressive(-like) state) was higher in G1 (G1: 4.10, G2: 0.54), consistent with G1 participants being less likely to be in the depressive state. When energy landscapes were constructed for the 4 periods, the stable states included 000000 and 010101/010110/011110/111110 (depressive-like states) for G1 participants and 000000, 010101, and 111111 for G2 participants (Figs 3G, 3H, and S6D–S6F). Note that G1 participants had a very small probability of being in state 111111 (fully depressive), and thus 111111 was not a stable state; either K6-1 or K6-6 was 0 in the stable state (i.e., their emotional state lacked nervousness or feelings of worthlessness). The energy of 000000 is 0 as defined in the model. The energies of the depressive-like states in G1 peaked in Period 2 and then decreased (3.57, 4.57, 3.60, and 3.76), but in G2 were at about the same level as 000000 (−0.39, 0.70, 0.50, and 0.72), indicating that G1 participants were least likely to exhibit depressive symptoms in Period 2, whereas G2 participants were equally likely to be healthy or depressed over the 4 periods.

To examine how G1 and G2 participants moved between different states over time, we performed simulations of state transitions constrained by the energy landscape of G1 or G2. Two simulations showed an interesting difference: In the G1 landscape, the mean time from 000000 to 111111 increased more than 2-fold in Periods 2–4 compared with Period 1 (Fig 3I), but the mean time from 111111 to 000000 did not differ significantly across the four periods (Fig 3J). In contrast, in the G2 landscape, the mean time from 000000 to 111111 did not significantly differ across the four periods (Fig 3K), whereas the mean time from 111111 to 000000 decreased more than 2-fold in Periods 2–4 compared with Period 1 (Fig 3L). These results suggest that the pandemic states of emergency prevented the G1 participants from becoming depressed, and at the same time triggered the G2 participants to move into a healthy state.

Again, this trend in the simulations was also confirmed by a statistical analysis of the two basins (the healthy 000000 basin and the depressive 111110(G1)/111111(G2) basin) (S13C-F Fig). The probability of staying in the healthy basin increased in Periods 2–4 compared to Period 1 in both G1 and G2 (S13C-D Fig), and the monthly transition probability from the healthy basin to the depressive basin decreased in G1 (S13E Fig), whereas the monthly transition probability from the depressive basin to the healthy basin increased in G2 (S13F Fig).

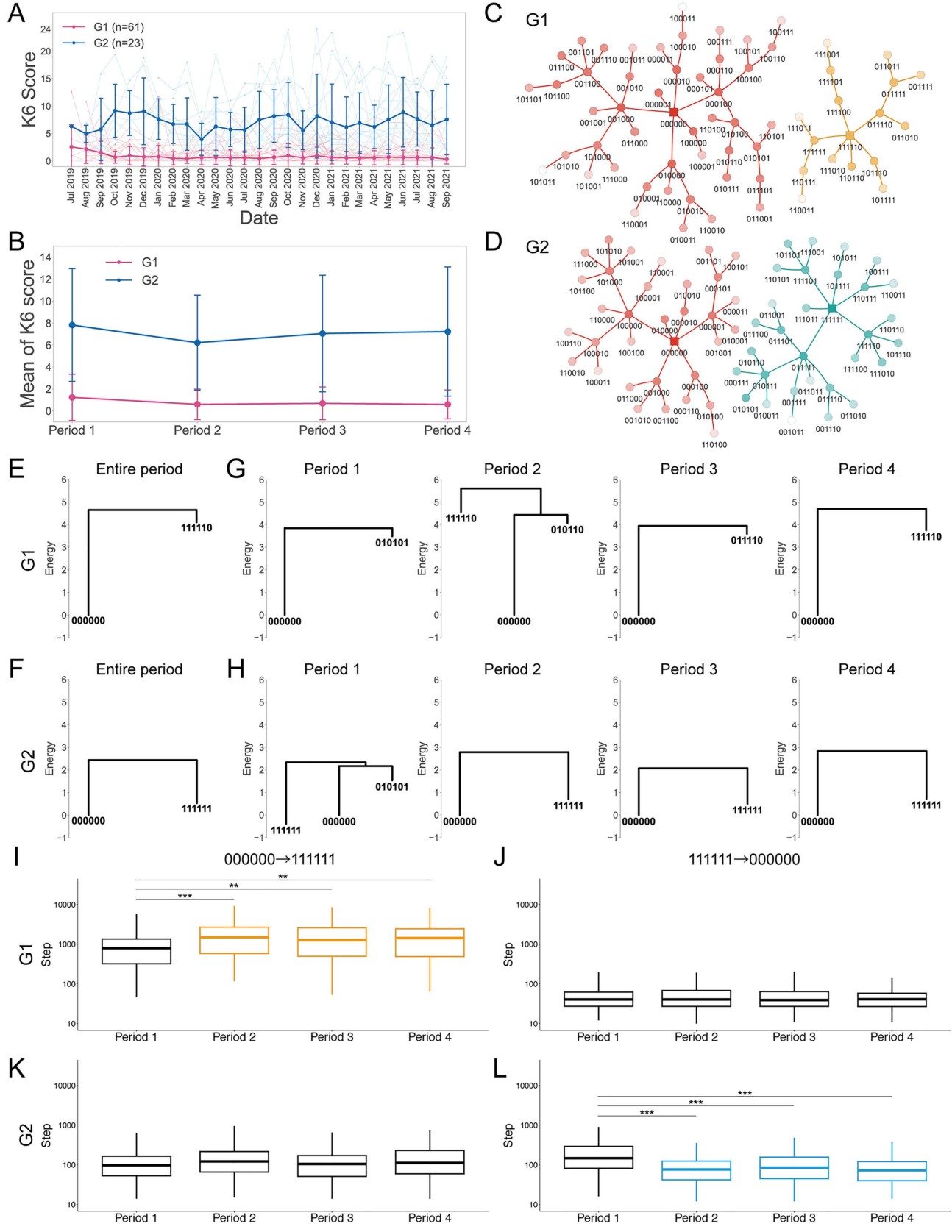

**Fig 3. Energy landscape analysis in the stratified groups. (A)** Clustering of time-series total K6 scores. Total K6 scores are plotted with time for each participant. The thick line represents the mean. Error bars indicate standard deviation (SD). The colors correspond to the stratified groups G1 and G2. G1 (plotted in pink) tended to have lower and more stable total K6 scores, whereas G2 (plotted in blue) tended to have higher and more variable scores. **(B)** The mean of the total K6 score for G1 and G2 is shown over the 4 periods. Periods 1–4, defined in Fig 2D, are as follows: Period 1 (July–December 2019), before the COVID-19 pandemic; Period 2 (January–May 2020), including the first COVID-19 case in Japan and the first state of emergency; Period 3 (June–December 2020), when no state of emergency was declared; and Period 4 (January–September 2021), including the second to fourth state of emergency declarations. Error bars indicate standard deviation (SD). **(C, D)** The graphs of the basin of attraction are drawn separately for G1 and G2 participants. **(E, F)** The disconnectivity graphs of the energy landscape for G1 and G2 are shown for the entire period. **(G, H)** The disconnectivity graphs of the energy landscape for G1 and G2 are shown for the 4 periods. **(I–L)** The simulated times from 000000 to 111111 or from 111111 to 000000 for G1 and G2 are compared between the 4 periods, respectively. 000000 (binarized responses of 0 to all six items) can be considered "healthy," whereas 111111 (binarized responses of 1 to all six items) represents the "depressive" state. Box plots show the median (center line), interquartile range (box), and range (whiskers). The upper outliers are hidden to improve visibility. Yellow and blue indicate increase and decrease, respectively, compared with Period 1.

## Biological characterization of diverse psychological responses to the COVID-19 pandemic

We further tested whether brain structural features and adolescent developmental trajectories differed between the G1 and G2 groups. In the pn-TTC, a total of 479 participants underwent brain MRI scans, of which a subset (84 participants) participated in a monthly web-based questionnaire survey (Figs 4A and S1 and S2 Methods). After modeling nonlinear developmental trajectories of each brain feature using GAMMs with participant-level random effects and extracting individual deviations from these models, GLMM analyses revealed a decreasing trend in the cortical thickness of the caudal part of the middle frontal gyrus (cMFG) during adolescence, and the degree of decrease was greater in G2 than in G1 after false discovery rate (FDR) correction ($t = -3.83$, $p = 0.00016$, $q = 0.012$; Fig 4C). On the other hand, the increase in GHQ-28 scores between Waves 3 and 4 was associated with an increase in cortical thickness in the temporal pole (TP) and entorhinal cortex ($t = 3.78$, $p = 0.00017$, $q = 0.0061$; $t = 3.75$, $p = 0.00019$, $q = 0.0061$ (FDR correction); Fig 4H) and a decrease in surface area in the same regions ($t = -3.15$, $p = 0.0017$, $q = 0.032$; $t = 3.69$, $p = 0.00024$, $q = 0.0061$ (FDR correction)). There was no significant association in the GHQ-28 score at Wave 3 (Fig 4E and 4F). For the three psychological features (energy-landscape-related G1/G2 clustering based on the K6 items, GHQ-28 scores at Wave 3, and difference in the GHQ-28 scores between Waves 3 and 4), there was no significant model for the main effect for any brain features.

We next tested whether the G1 or G2 group or the difference in GHQ-28 scores was more significantly associated with change in five brain features (cortical thickness of cMFG, cortical thickness and surface area of entorhinal cortex, and cortical thickness and surface area of TP) by simultaneously adding the psychological features to the GLMMs. The models showed significant age × group interactions, but not age × GHQ-28 score difference interactions, on the cortical thickness of cMFG (Figs 4B and S18 and S8 Table and S5 Note; $t = -2.36$, $p = 0.019$, $q = 0.048$ (FDR correction); $t = -0.57$, $p = 0.57$) and TP (Figs 4B and S18 and S9 Table and S5 Note; $t = 3.08$, $p = 0.0023$, $q = 0.012$ (FDR correction); $t = -1.03$, $p = 0.31$), indicating the direction of accelerated brain maturation in G2 compared with G1.

Taken together, the energy-landscape-related group was associated with the change in the cortical thickness in the cMFG and TP, with G2 (the group with higher and unstable K6 scores) showing a greater change. This suggests that morphological changes in these brain regions may be associated with increased sensitivity to psychological fluctuations.

## Discussion

In this study, we examined monthly questionnaire responses by adolescents in Japan about their psychological distress and analyzed the effect of states of emergency during the COVID-19 pandemic. Specifically, our longitudinal study included periods both before and during the pandemic (including the declarations of states of emergency), allowing us to directly assess the effects of the states of emergency. By dividing the study dataset into four time periods and applying energy landscape analysis to each, we examined the temporal aspect of psychological distress. Comparison of disconnectivity graphs before

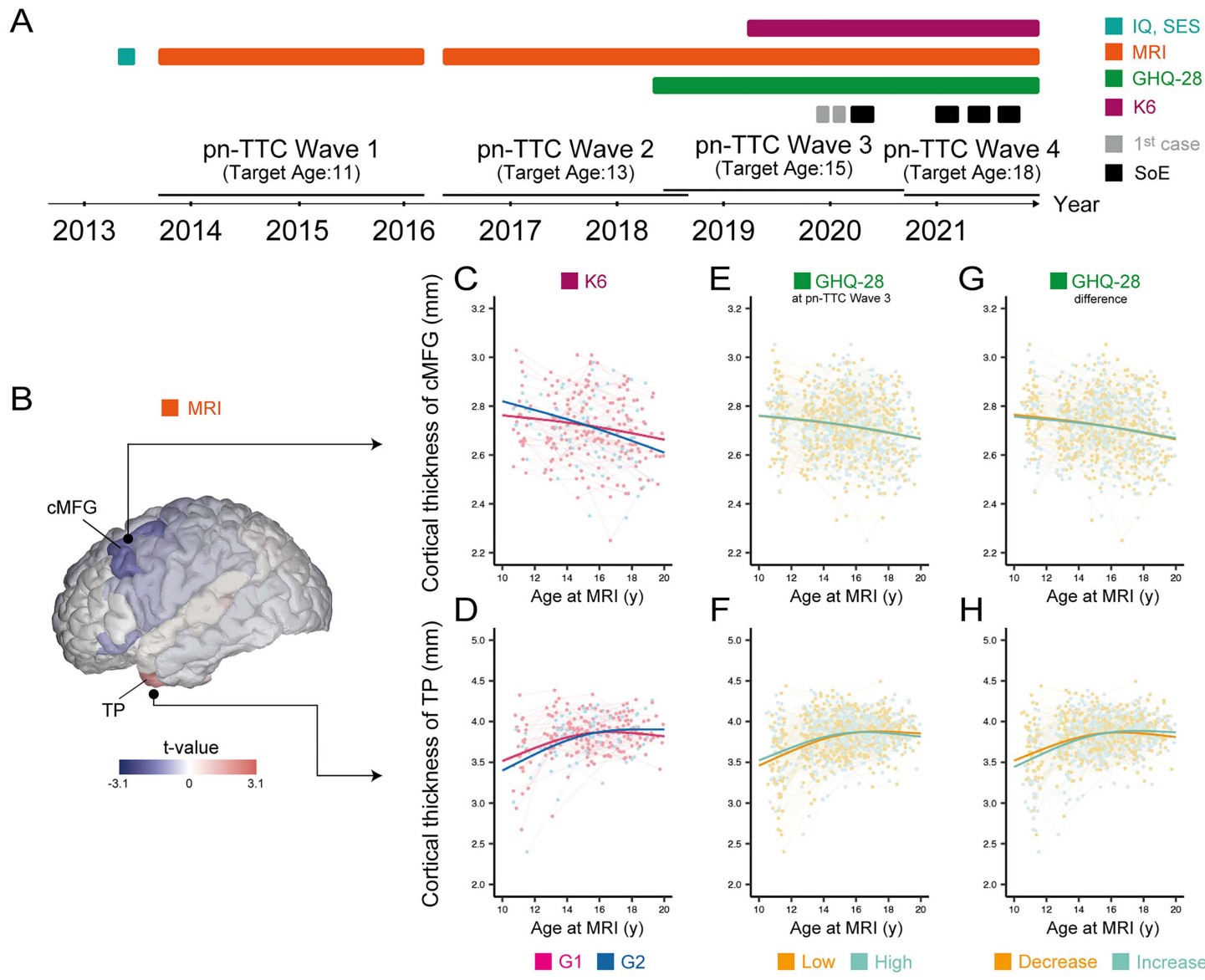

**Fig 4. Developmental trajectories of brain features in stratified groups. (A)** Summary of data collection schedule (see also Methods, Figs 2D and S1 and S1 Methods). IQ, intelligence quotient; SES, socioeconomic status; SoE, state of emergency. **(B)** cMFG and TP in the brain cortex are visualized along with *t*-values in the GLMM analysis (S8 and S9 Tables). cMFG, caudal part of the middle frontal gyrus; TP, temporal pole. **(C–H)** Developmental trajectories of brain features for each group are plotted with age at MRI (y) on the *x*-axis and cortical thickness on the y-axis for each participant. The thick line represents the estimated value by general linear mixed models (GLMMs). **(C)** A greater decrease in the cortical thickness of cMFG is observed in G2 than in G1. G1 and G2 represent the clustering groups identified in Fig 3; G1 showed lower and more stable total K6 scores, whereas G2 showed higher and more variable scores. **(D)** A greater increase in the cortical thickness of TP is observed in G2 than in G1. **(E, F)** Cortical thickness of cMFG and TP in 479 participants, colored by GHQ-28 score (low/high) at Wave 3. **(G, H)** Cortical thickness of cMFG and TP in 479 participants, colored by the difference in GHQ-28 score (decrease/increase) between Waves 3 and 4. The illustration of the human brain in Fig 4 was drawn using BrainPainter (https://github.com/razvanmarinescu/brain-coloring).

and after the pandemic revealed an association between the pandemic period and a reduced likelihood of psychological distress. Numerical simulations of the energy landscape also suggested that the pandemic states of emergency were associated with a reduced probability of transitions from healthy states to depressive states.

Our findings are consistent with reports from the United Kingdom [45], China [11,46], and South Korea [47,48], which indicate an improvement in the mental health of adolescents in response to the COVID-19 public health emergency. However, our results differ from reports from the United States [49] and Europe [50,51], which suggested a general decline in adolescents' mental health during the pandemic. These seemingly contradictory findings point to two different aspects of high school. In one aspect, schools are a fun place that provide face-to-face interaction with peers; the pandemic deprived students of such opportunities. In another aspect, schools are a nuisance, forcing students to participate in mandatory activities and study in preparation for college entrance exams (especially in East Asia, including Japan); the pandemic and school closures freed students from this, if only temporarily. These results also highlight the variability in the resilience of adolescents in the face of the pandemic [52]. Alternatively, external factors, such as family socioeconomic status, parental stress, and access to the internet including social media, may have played a role [53,54]. In fact, our participants tended to come from relatively affluent families who were cooperative with our repeated surveys; therefore, it is possible that the adolescents were content to stay home rather than go to school.

Our analysis successfully identified two groups of students with different psychological characteristics, with G1 having lower and more stable total K6 scores and G2 having higher and more volatile total K6 scores. In particular, both numerical simulations and statistical analyses of basins suggested that during the pandemic, G1 participants were less likely to switch from healthy to depressive states, whereas G2 participants were more likely to switch from depressive to healthy states. Notably, brain MRI analysis showed that G2 participants had greater changes in the cortical thickness of the cMFG and TP than did G1 participants, and the effect was greater than the difference in depressive symptoms assessed on the dates of the MRI scans. The changes were in the direction of accelerated adolescent brain development, suggesting that morphological changes in these regions, which are responsible for higher-order cognitive functions, may be associated with increasing sensitivity to psychological fluctuations.

Our observation that the state of emergency shifted G2 participants from a depressive state to a healthy state suggests that these participants may have returned to the depressive state when the state of emergency was lifted. In fact, in our analysis, the energy of 111,111 was slightly lower in Period 4 than in Periods 2 and 3 (Fig 2F), suggesting that the probability of being in the depressive state increased again in Period 4 (see also S13A Fig). This is reminiscent of reports from Japan [55,56] that the later phase of the pandemic, when schools reopened, saw an increase in the incidence of suicide.

What can we learn from these findings to prepare for the next pandemic? Our results revealed an unexpected degree of variability not only in the frequency of psychological states, but also in their transition dynamics. In this sense, our work can be placed in the lineage of recent studies that view health and disease in the context of dynamical systems [57–60]. Dynamical systems theory may help not only to understand disease states, but also to intervene and prevent them by detecting early warning signals [61,62]. Disasters such as pandemics are particularly suited to this type of analysis because the external cause is clearly defined and, although variability exists, individuals tend to have similar responses. For example, examining inter-item associations of K6 identified by the energy landscape analysis, beyond the total K6 score, could provide deeper insights into how specific symptoms, such as concentration difficulties, fatigue, or feelings of hopelessness, interact and evolve over time. These item-level patterns may highlight distinct psychological profiles that are overlooked by aggregate scoring, helping educators and school health professionals to identify at-risk students and tailor preventive support more effectively during future public health crises. Further characterization of individual variability will require more data, but it is clear that we need to be vigilant about the possible adverse effects of the next pandemic on adolescents and provide psychological support to those who need it. Furthermore, in addition to characterizing psychological dynamics, the energy landscape analysis could be integrated into economic and predictive modeling frameworks to quantify the potential value of early mental health interventions. For instance, by linking energy landscape-derived stability

metrics with longitudinal outcomes such as absenteeism, academic performance, or clinical referrals, future studies could estimate the cost-effectiveness of targeted school-based prevention under different resource or pandemic scenarios. This integration may ultimately inform policy decisions on allocating resources for youth mental health programs.

We acknowledge some limitations of this study. First, the participants in this cohort lived in Tokyo only, and the generalizability of these findings to adolescents living in other local areas and countries needs to be confirmed. Future studies that include energy landscape analysis for participants of different demographic backgrounds, such as other age groups, would provide a useful perspective for earlier prediction of depressive symptoms. Second, recruitment of participants to the monthly questionnaire survey took place on the date of MRI imaging, and the number of participants gradually increased from before the pandemic to the state of emergency. Although this prevented exact individual comparisons between different periods, our method enabled analysis by extracting dynamics from the pooled population data. Careful design in future cohorts will allow for more precise assessments through individual comparisons. In addition, a larger sample size and longer time period may provide a fluctuation pattern independent of specific events. Smartphone apps and wearable devices could be further utilized to incorporate Ecological Momentary Assessment to assess participants' psychological states in detail [63]. Third, the current energy landscape analysis required the data to be binarized, in part due to the combinatorial explosion of the number of possible states. In our analysis, this simplified and emphasized the contrast between healthy and depressive states, but it is possible that removing the binarization may reveal other "sych'logical states. Furthermore, while the current energy landscape analysis relies on the Markovian assumption, future methodological extensions could relax this constraint to better capture temporal dependencies inherent in longitudinal psychological processes. For example, adopting a kinetic Ising-based formulation or incorporating lagged variables may enable the model to represent non-Markovian dynamics and memory effects more realistically. Fourth, the present study was not designed to perform predictive modeling or cross-validation analyses. Accordingly, the robustness of the identified energy landscape structures in terms of out-of-sample prediction was not formally evaluated and should be examined in future studies, ideally using independent cohorts. Finally, the association we found between psychological states and brain development remains descriptive, and further experimental or interventional studies are needed to validate causality.

Energy landscape analysis has been successfully applied to chemical physics [64], neuroscience [19,65–67], and ecology [35,68]. Energy-based explanations of multistable behavior help to extract temporal information, even from short-item questionnaires such as the K6, and are a useful complement to conventional statistical analyses. This work further extends the repertoire of energy landscape analysis to the fields of psychology and psychiatry, where multidimensional questionnaire-based surveys abound and the concept of dynamical systems is effective for explaining disease states [57,62]. Future studies should test the model's generalizability by applying the energy landscape analysis to other cohorts, thereby evaluating its predictive validity and practical relevance across diverse populations. Such efforts will surely lead to unraveling the evolving landscape of many more psychological and psychiatric situations.

## Supporting information

**S1 Fig. Summary of the whole study.**
(DOCX)

**S2 Fig. Preprocessing of each participant's questionnaire responses.**
(DOCX)

**S3 Fig. Schematic illustration of energy landscape analysis on the K6 questionnaire.**
(DOCX)

**S4 Fig. Different smoothing methods.**
(DOCX)

 

**S5 Fig. Different binarization methods.**
(DOCX)

**S6 Fig. Accuracy of the model fitting.**
(DOCX)

**S7 Fig. Confidence intervals of parameter estimate.**
(DOCX)

**S8 Fig. Convergence of estimation errors and changes in parameter estimates across iterations.**
(DOCX)

**S9 Fig. Changes in total K6 scores across different periods.**
(DOCX)

**S10 Fig. Relationship between school days per week and K6 questionnaire responses.**
(DOCX)

**S11 Fig. Latent class analysis of time-series K6 questionnaire responses for all participants.**
(DOCX)

**S12 Fig. The average dwell time of each state in the simulation.**
(DOCX)

**S13 Fig. Probabilities of staying in the basin and of transitioning between basins.**
(DOCX)

**S14 Fig. Energy landscape analysis of time-series K6 questionnaire responses by month.**
(DOCX)

**S15 Fig. Confidence intervals of energy estimates.**
(DOCX)

**S16 Fig. Clustering of participants by dynamic time warping.**
(DOCX)

**S17 Fig. Accuracy of the clustering.**
(DOCX)

**S18 Fig. Estimated power at a range of sample sizes.**
(DOCX)

**S1 Note. Glossary of energy landscape analysis terms.**
(DOCX)

**S2 Note. Comparison of different binarization and smoothing methods.**
(DOCX)

**S3 Note. Validation of energy landscape analysis using latent class analysis.**
(DOCX)

**S4 Note. Further statistical analysis on probabilities of staying in the basin and of transitioning between basins.**
(DOCX)

**S5 Note. Power calculation.**
(DOCX)

**S1 Methods. Data acquisition of population-neuroscience Tokyo TEEN Cohort.**
(DOCX)

**S2 Methods. Acquisition and preprocessing methods for the brain imaging procedures.**
(DOCX)

**S1 Table. Image acquisition parameters and preprocessing methods.**
(DOCX)

**S2 Table. Difference in depressive symptom scores between energy-landscape-related G1 and G2 groups.**
(DOCX)

**S3 Table. List of the model variables and parameters.**
(DOCX)

**S4 Table. Estimated coefficients of the Ising model for the entire period.**
(DOCX)

**S5 Table. Item-level statistical analysis.**
(DOCX)

**S6 Table. Number of participants, categorized by the mean total K6 scores, in each Period.**
(DOCX)

**S7 Table. Demographic characteristics in energy-landscape-related G1 and G2 groups.**
(DOCX)

**S8 Table. Relationship of three psychological features with the cortical thickness in the caudal part of the middle frontal gyrus (cMFG).**
(DOCX)

**S9 Table. Relationship of three psychological features with the cortical thickness in the temporal pole (TP).**
(DOCX)

**S1 Checklist. STROBE Checklist.** STrengthening the Reporting of OBservational studies in Epidemiology (STROBE) Statement—checklist of items that should be included in reports of observational studies, available at https://www. strobe-statement.org/, licenced under CC BY 4.0.
(DOCX)

**S2 Checklist. TRIPOD-AI Checklist.**
(DOCX)

## Author contributions

**Conceptualization:** Naotoshi Nakamura, Shinsuke Koike, Shingo Iwami.

**Data curation:** Lin Cai, Atsushi Nishida, Naohiro Okada, Kiyoto Kasai, Shinsuke Koike.

**Formal analysis:** Daiki Tatematsu, Shinsuke Koike.

**Funding acquisition:** Kazuyuki Aihara, Kiyoto Kasai, Shinsuke Koike, Shingo Iwami.

**Investigation:** Daiki Tatematsu, Naotoshi Nakamura, Lin Cai, Shingo Iwami.

**Methodology:** Masato S. Abe, Tetsuo Ishikawa, Takahiro Ezaki, Lin Cai, Eiryo Kawakami, Naoki Masuda, Kiyoto Kasai.

**Project administration:** Kazuyuki Aihara, Atsushi Nishida, Kiyoto Kasai, Shinsuke Koike, Shingo Iwami.

**Resources:** Masato S. Abe, Tetsuo Ishikawa, Takahiro Ezaki, Lin Cai, Atsushi Nishida, Naohiro Okada, Kiyoto Kasai, Shinsuke Koike.

**Software:** Daiki Tatematsu, Takahiro Ezaki.

**Supervision:** Naotoshi Nakamura, Shinsuke Koike, Shingo Iwami.

**Validation:** Daiki Tatematsu, Naotoshi Nakamura, Atsushi Nishida, Naohiro Okada, Shinsuke Koike.

**Visualization:** Daiki Tatematsu, Naotoshi Nakamura, Shinsuke Koike.

**Writing – original draft:** Daiki Tatematsu, Naotoshi Nakamura, Shinsuke Koike, Shingo Iwami.

**Writing – review & editing:** Daiki Tatematsu, Naotoshi Nakamura, Masato S. Abe, Tetsuo Ishikawa, Takahiro Ezaki, Lin Cai, Eiryo Kawakami, Kazuyuki Aihara, Atsushi Nishida, Naohiro Okada, Naoki Masuda, Kiyoto Kasai, Shinsuke Koike, Shingo Iwami.

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
