## [Editor Report · Decision Letter 0]

19 Sep 2024

Dear Dr Iwami,

Thank you for submitting your manuscript entitled "Evolving landscape of psychological distress among high school students during the COVID-19 pandemic" for consideration by PLOS Medicine.

Your manuscript has now been evaluated by the PLOS Medicine editorial staff and I am writing to let you know that we would like to send your submission out for external peer review.

Please re-submit your manuscript within two working days, i.e. by Sep 23 2024.

Feel free to email me at atosun@plos.org or us at plosmedicine@plos.org if you have any queries relating to your submission.

Kind regards,

Alexandra Tosun, PhD

Associate Editor

PLOS Medicine

---

## [Decision Letter · Decision Letter 1]

25 Apr 2025

Dear Dr Iwami,

Many thanks for submitting your manuscript "Evolving landscape of psychological distress among high school students during the COVID-19 pandemic" (PMEDICINE-D-24-03134R1) to PLOS Medicine. The paper has been reviewed by a subject expert and a statistician; their comments are included below and can also be accessed here: [LINK]

We have considered your appeal request and I am writing to let you know that we are willing to reconsider the manuscript. Please note that the manuscript will undergo further external review and that we cannot provide any guarantees at this stage regarding publication.

We ask that you submit your revision by May 16 2025. However, if this deadline is not feasible, please contact me by email, and we can discuss a suitable alternative.

Don't hesitate to contact me directly with any questions (atosun@plos.org).

Best regards,

Alexandra

Alexandra Tosun, PhD

Associate Editor

PLOS Medicine

atosun@plos.org

Comments from the reviewers:

Reviewer #1: Important study, but there are some major points to clarify the study.

Abstract

Please indicate the number of samples (total and important subgroups), and K6 cut-offs.

Throughout the study

K6 cut-offs can significantly change the interpretation of the results of this analysis. Please clearly mention how cut-offs of K6 (5, 8 or 13) are used to determine the severity of the distresses in this study.

In relation to the above cut-off value of K6, it is difficult to understand the explanation of G1 and G2. Please explain carefully what is meant by higher K6 or low K6.

Discussion

It is difficult to understand the overall picture of what happened to what percentage of people (total and subgroups) in this analysis. Therefore, it is not clear how the results can be applied to future pandemics by what degree. There are some major limitations in applying this study to all of Japan and the world as a whole, but we hope that you will take this into consideration and improve this aspect in the text.

Reviewer #2: 1) The manuscript uses the Ising model in physics as the basis for the energy landscape analysis, which assigns energies to states representing responses to the K6 items. However, there is no detailed explanation about why the Ising model was selected over alternative models in the context of psychological data.

2) Binarization of Data: The conversion of continuous K6 scores into binary data (above or below the mean) may lead to a loss of information. The authors should discuss the limitations of binarizing continuous data and its impact on the sensitivity of the results. Additionally, exploring how different thresholds (such as median or clinically relevant cut-offs) impact the energy landscape would be valuable.

3) The manuscript mentions the use of two accuracy indices—entropy and Kullback-Leibler divergence—to validate the pairwise maximum entropy model. These are appropriate choices, but it would be useful to include a more in-depth comparison of the energy landscape model with traditional statistical models used in psychology to demonstrate its added value.

4) The authors acknowledge that constructing energy landscapes based on monthly data may lead to lower accuracy due to a small sample size. Including statistical diagnostics for this potential limitation, such as confidence intervals around the energy estimates, would enhance the reader's understanding of the robustness of the findings.

5) The manuscript uses Gibbs sampling to simulate transitions between the depressive (111111) and healthy (000000) states. While the results are clear, the assumptions behind the transition process are not fully explained. For instance, does the energy landscape assume Markovian properties, and if so, how is the memoryless nature of transitions justified for psychological data, which may show temporal dependence?

6) In some sections, the median time for transitions between states is compared across periods. A more statistical comparison between transition times (e.g., confidence intervals, hypothesis testing, or bootstrapping) would provide more rigor to these comparisons.

7) The manuscript stratifies participants into two groups (G1 and G2) using k-means clustering based on their K6 scores. While k-means clustering is widely used, it might not always be the most appropriate method for clustering time-series data. The use of time-series-specific clustering methods (such as dynamic time warping) could be explored. At a minimum, the authors should discuss why k-means was chosen and how sensitive the results are to the number of clusters.

8) The group-level comparisons could be expanded with additional statistical tests, such as ANOVA or mixed-effects models, to provide more robust insights into the differences between the groups in terms of their psychological distress and brain morphology.

9) The mathematical equations for the maximum entropy model and the computation of probabilities using the energy function are clearly explained. However, readers who are not familiar with the Ising model might struggle with the concepts. It would be helpful if the authors provided an intuitive example or additional explanation of how the model handles interactions between different psychological items (e.g., K6-1 to K6-6).

10) The inclusion of pairwise interaction terms (Jij) between items in the model is briefly mentioned, but the implications of these interactions should be discussed in more detail. Are the interactions assumed to be symmetric, and how are they estimated from the data?

11) As noted earlier, the binarization process could significantly impact the model outcomes. A sensitivity analysis exploring how the results vary when different thresholds are used (e.g., different cut-offs for binarization or alternative scoring methods) would improve the robustness of the findings.

12) Although the energy landscape analysis is the focus of the manuscript, comparing its performance to more traditional statistical models (e.g., latent class analysis or longitudinal mixed-effects models) would provide context for the added value of ELA.

13) The manuscript links psychological distress profiles (from energy landscape analysis) with changes in cortical thickness. This is an interesting application, but more details are needed on how these associations were tested. Were the brain data adjusted for potential confounders such as age, sex, and socioeconomic status? Additionally, multiple comparison correction methods should be clearly stated given the number of brain regions analysed.

14) Longitudinal studies often face challenges with missing data or participant dropout over time. The manuscript should discuss how missing data were handled, especially since the data collection spanned over two years. If imputation methods were used, details about the method and its justification are essential. Alternatively, a discussion on how the missing data could have impacted the results (e.g., selection bias or reduced statistical power) should be included.

15) While the manuscript mentions the use of Gaussian process regression to smooth time-series data, it would be beneficial to explain why this approach was chosen over others (such as Kalman filtering or spline-based methods) for handling irregular time points. Additionally, clarifying how the temporal aspect of psychological distress is accounted for in the energy landscape framework would be helpful, as this is crucial in longitudinal data.

16) The manuscript presents multiple complex statistical outputs, including energy landscapes, transition times, and brain structure associations. While each result is explained in isolation, the overall narrative could be improved by integrating these findings more clearly. For instance, the discussion could benefit from a summary that ties together how the psychological states (as modelled by the energy landscape) correlate with brain structural changes and how these dynamics evolve throughout the pandemic.

17) The study infers that the COVID-19 pandemic and the associated states of emergency impacted psychological states and brain development. However, the ability to make causal claims from the energy landscape model, which is fundamentally descriptive, should be carefully nuanced. The manuscript should acknowledge that causal interpretations are limited unless supported by further experimental or interventional data.

18) In sections where multiple brain regions or psychological states are being analysed, it is crucial to ensure that appropriate corrections for multiple comparisons (such as Bonferroni correction or False Discovery Rate) are applied. While the manuscript references some corrections, it would be beneficial to explicitly describe which corrections were applied and to what analyses.

19) There is no mention of power calculations in the manuscript. Given the relatively small sample size (84 participants), especially when analysing brain imaging data, a power analysis would help to assess whether the study had sufficient power to detect meaningful differences in psychological distress or brain structural changes. This is particularly relevant for detecting subtle effects in longitudinal psychological and brain measures.

20) The binarization process simplifies continuous psychological distress scores into "above or below average" categories. This simplification may lead to the loss of important granularity. The discussion should elaborate on what is lost or gained by this binning approach, and whether any important psychological nuances are potentially missed by treating all individuals above or below the threshold as homogeneous groups.

21) The energy landscape model treats each state (combination of K6 scores) as either "healthy" or "depressive," but this binary treatment of psychological distress might overlook intermediary states of mental health. Providing more interpretation of what the middle ground looks like (e.g., scores that are neither fully healthy nor fully depressive) would help bridge the gap between the model and clinical reality.

22) The novel nature of the energy landscape analysis approach may make it harder for other researchers to replicate. The manuscript should include clear instructions or supplementary material with code to allow others to reproduce the results. If any custom software or scripts were used for the energy landscape modelling or Gibbs sampling simulations, these should be shared in a public repository.

23) It is important to report diagnostic checks for the energy landscape models. For example, did the model converge properly? Were any diagnostics conducted to ensure that the model fits the data appropriately? Reporting these checks would enhance the reader's confidence in the robustness of the analysis.

24) Beyond the two clusters identified (G1 and G2), it might be worth exploring whether other demographic variables (e.g., gender, socioeconomic status, academic performance) could serve as additional or alternative stratification factors. Some subgroups may have experienced different psychological impacts during the pandemic, and the current stratification based purely on K6 scores might overlook these effects.

---

* Please upload any figures associated with your paper as individual TIF or EPS files with 300dpi resolution at resubmission; please read our figure guidelines for more information on our requirements: http://journals.plos.org/plosmedicine/s/figures. While revising your submission, please upload your figure files to the PACE digital diagnostic tool, https://pacev2.apexcovantage.com/. PACE helps ensure that figures meet PLOS requirements. To use PACE, you must first register as a user. Then, login and navigate to the UPLOAD tab, where you will find detailed instructions on how to use the tool. If you encounter any issues or have any questions when using PACE, please email us at PLOSMedicine@plos.org.

* ETHICS STATEMENT: Please indicate whether consent was waived.

* FINANCIAL DISCLOSURES: The funding statement should include: specific grant numbers, initials of authors who received each award, URLs to sponsors’ websites. Also, please state whether any sponsors or funders (other than the named authors) played any role in study design, data collection and analysis, the decision to publish, or preparation of the manuscript. If they had no role in the research, include this sentence: “The funders had no role in study design, data collection and analysis, decision to publish, or preparation of the manuscript.”

* DATA AVAILABILITY STATEMENT: PLOS Medicine requires that the de-identified data underlying the specific results in a published article be made available, without restrictions on access, in a public repository or as Supporting Information at the time of article publication, provided it is legal and ethical to do so. Please see the policy at http://journals.plos.org/plosmedicine/s/data-availability and FAQs at

http://journals.plos.org/plosmedicine/s/data-availability#loc-faqs-for-data-policy

* DATA AVAILABILITY STATEMENT: The Data Availability Statement (DAS) requires revision. For each data source used in your study:

FIGURES AND TABLES

SUPPLEMENTARY MATERIAL

REFERENCES

STUDY TYPE-SPECIFIC REQUESTS

* Abstract: Please include the study design, population and setting, number of participants, years during which the study took place (enrollment and follow up), length of follow up, and main outcome measures.

* If available, please ensure that the study is reported according to an appropriate guideline, such as STROBE (available from: https://www.equator-network.org/reporting-guidelines/strobe) and include the completed checklist as Supporting Information.

* In the manuscript text, please indicate: (1) the specific hypotheses you intended to test, (2) the analytical methods by which you planned to test them, (3) the analyses you actually performed, and (4) when reported analyses differ from those that were planned, transparent explanations for differences that affect the reliability of the study's results. If a reported analysis was performed based on an interesting but unanticipated pattern in the data, please be clear that the analysis was data driven.

* Please state in the Methods section whether the study had a prospective protocol or analysis plan. If a prospective analysis plan (from your funding proposal, IRB or other ethics committee submission, study protocol, or other planning document written before analyzing the data) was used in designing the study, please include the relevant document(s) with your revised manuscript as a Supporting Information file to be published alongside your study and cite it in the Methods section. A legend for this file should be included at the end of your manuscript. If no such document exists, please make sure that the Methods section transparently describes when analyses were planned, and when/why any data-driven changes to analyses took place. Changes in the analysis, including those made in response to peer review comments, should be identified as such in the Methods section of the paper, with rationale.

* The following list is derived from Geoffrey P Garnett, Simon Cousens, Timothy B Hallett, Richard Steketee, Neff Walker. Mathematical models in the evaluation of health programmes. (2011) Lancet DOI:10.1016/S0140-6736(10)61505-X:

* If pertinent, please provide a diagram that shows the model structure, including how the natural history of the disease is represented, the process and determinants of disease acquisition, and how the putative intervention could affect the system.

* Please provide a complete list of model parameters, including clear and precise descriptions of the meaning of each parameter, together with the values or ranges for each, with justification or the primary source cited and important caveats about the use of these values noted.

* Please provide a clear statement about how the model was fitted to the data, including goodness-of-fit measure, the numerical algorithm used, which parameter varied, constraints imposed on parameter values, and starting conditions.

* For uncertainty analyses, please state the sources of uncertainties quantified and not quantified [can include parameter, data, and model structure].

* Please provide sensitivity analyses to identify which parameter values are most important in the model. Uncertainty estimates seek to derive a range of credible results on the basis of an exploration of the range of reasonable parameter values. The choice of method should be presented and justified.

* Please discuss the scientific rationale for the choice of model structure and identify points where this choice could influence conclusions drawn. Please also describe the strength of the scientific basis underlying the key model assumptions.

* For studies that develop a prediction model or evaluate its performance, please ensure that the study is reported according to the TRIPOD statement (https://www.equator-network.org/reporting-guidelines/tripod-statement) and include the completed checklist as Supporting Information. Please add the following statement, or similar, to the Methods: "This study is reported as per the Transparent Reporting of a Multivariable Prediction Model for Individual Prognosis Or Diagnosis (TRIPOD) statement (S1 Checklist)." For studies using machine learning, please use the TRIPOD-AI checklist. When completing the checklist, please use section and paragraph numbers, rather than page numbers.

---

## [Decision Letter · Decision Letter 2]

8 Oct 2025

Dear Dr. Iwami,

Thank you very much for re-submitting your manuscript "Evolving landscape of psychological distress among Japanese high school students during the COVID-19 pandemic" (PMEDICINE-D-24-03134R2) for review by PLOS Medicine.

First, we would like to apologize for the unusually long time it has taken us to provide you with a decision. Unfortunately, we were unable to secure an additional subject review. We are unsure of the reasons behind this. One reviewer who initially agreed to review your manuscript did not deliver a review. We greatly appreciate your patience throughout this process, and we are glad to finally provide you with a decision.

We'd also like to thank you for your detailed response to the reviewers' and editors’ comments. I have discussed the paper with my colleagues, and it has also been seen again by the two original reviewers. The changes made to the paper were mostly satisfactory to the reviewers. The statistical reviewer has raised several points to further strengthen the paper. Please address the reviewers' and editors' comments below in a further revision. When submitting your revised paper, please once again include a detailed point-by-point response to the reviewers’ and editorial comments.

In revising the manuscript for further consideration here, please ensure you address the specific points made by each reviewer and the editors. In your rebuttal letter you should indicate your response to the reviewers' and editors' comments and the changes you have made in the manuscript. Please submit a clean version of the paper as the main article file. A version with changes marked must also be uploaded as a marked up manuscript file. Please also check the guidelines for revised papers at http://journals.plos.org/plosmedicine/s/revising-your-manuscript for any that apply to your paper.

We ask that you submit your revision within 1 week (Oct 15 2025). However, if this deadline is not feasible, please contact me by email, and we can discuss a suitable alternative.

Please do not hesitate to contact me directly with any questions (atosun@plos.org).

We look forward to receiving the revised manuscript.

Sincerely,

Alexandra Tosun, PhD

Senior Editor

PLOS Medicine

plosmedicine.org

Comments from Reviewers:

Reviewer #1: I have carefully reviewed the response letter and the revised manuscript. It is clear that the authors have appropriately and thoroughly addressed the reviewer comments. The revisions reflect a sincere effort to enhance clarity, methodological transparency, and accessibility of the energy landscape analysis. The additional explanations, sensitivity analyses, and improved figure annotations contribute meaningfully to the overall quality and reproducibility of the work. I believe the authors have responded in good faith to all points raised, and no major concerns remain unaddressed.

Reviewer #2: I appreciate the authors' thorough revisions and the clear effort made to address my previous comments. The manuscript is substantially improved. I would like to offer a few further suggestions to enhance the statistical and methodological robustness of the analysis.

a) While confidence intervals around energy estimates are provided for selected states, consider extending uncertainty quantification to all estimated parameters of the Ising model (e.g., hih_ihi, JijJ_{ij}Jij) to allow a more comprehensive assessment of robustness, especially across time periods.

b) Although the manuscript mentions validation via entropy and KL divergence, it would be helpful to include additional information on model convergence diagnostics (e.g., likelihood convergence plots, changes in parameter estimates over iterations) to support the reliability of the energy landscape estimates.

c) While the authors acknowledged the Markovian assumption and its limitations, exploring even a basic extension to non-Markovian transitions (e.g., via lagged variables or conditional entropy) could improve the model's realism for longitudinal psychological data. At minimum, this should be acknowledged as a potential future improvement in the Discussion.

d) The current model is descriptive, and while it shows intuitive trends, some form of cross-validation (e.g., time-based splitting) or out-of-sample predictive accuracy (for transitions or group classification) could help establish its generalisability and utility in future applications.

e) While the energy landscape analysis has been contrasted descriptively with traditional methods, a quantitative comparison of predictive performance (e.g., using AIC/BIC, likelihood ratios, or classification metrics) between ELA and alternative models (like mixed-effects models or LCA) could better establish its empirical advantage.

f) While it is appreciated that the Python version of the modelling code is based on an existing MATLAB repository, I strongly recommend making the actual Python scripts used in this study available (e.g., via GitHub and archived on Zenodo) at the time of publication to meet reproducibility standards.

g) The Ising model's structure implies inter-item associations, yet their clinical interpretation remains underdeveloped. I suggest including a brief discussion of how these item-level interactions (e.g., between specific K6 items) might inform mental health screening or interventions in school settings.

h) Given the authors' novel use of energy landscapes, it may be helpful to briefly discuss the potential for integrating such models into economic evaluations or predictive modelling frameworks (e.g., for assessing cost-effectiveness of school-based interventions under pandemic scenarios). This could broaden the relevance of the approach.

Requests from Editors:

We would like to emphasize point e), made by the statistical reviewer, who noted that a quantitative comparison of the predictive performance of ELA and alternative models could better demonstrate its empirical advantage. We strongly agree and encourage you to include the suggested analysis.

GENERAL

* Please confirm that your title complies with to PLOS Medicine's style. Your title must be nondeclarative and not a question. It should begin with main concept if possible. "Effect of" should be used only if causality can be inferred, i.e., for an RCT. Please place the study design ("A randomized controlled trial," "A retrospective study," "A modelling study," etc.) in the subtitle (i.e., after a colon).

* Statistical reporting: Please revise throughout the manuscript, including tables and figures.

- Please report statistical information as follows to improve clarity for the reader ""XX (95% CI [XX,XX]; p</=)"".

- Please separate upper and lower bounds with commas instead of hyphens as the latter can be confused with reporting of negative values.

- Please repeat statistical definitions (HR, CI etc.) for each set of parentheses.

* Please ensure that all abbreviations are defined at first use throughout the text (including statistical abbreviations). Please also check figures and tables.

* Please ensure that tables and figures, including those in supplementary files, are appropriately referenced in the main text.

* Please check that any use of statistical terms (such as trend or significant) are supported by the data, and if not please remove them.

* Please ensure that where relevant figures include 95% CIs.

* Please review your text for claims of novelty or primacy (e.g. 'for the first time') and remove this language.

* Data availability: “The data will be available after a review process by the pn-TTC projects” – Does this mean that researchers interested in the data must submit a proposal via the email address you provided, which will then be reviewed by pn-TTC projects? Could you explain why the data is not publicly available?

ABSTRACT

* Please confirm that your abstract complies with our requirements, including providing all the information relevant to this study type https://journals.plos.org/plosmedicine/s/submission-guidelines#loc-abstract

* Please include the study setting.

* We feel that the Abstract Methods section is not sufficiently detailed. Given that most readers will be unfamiliar with the energy landscape analysis, please provide sufficient explanation.

* How is a healthy state and depressive state defined (scores)?

* Please quantify the main results (if applicable, with 95% CIs and p values).

* Please provide the main baseline characteristics of the study population.

* Did all participants undergo MRI?

* Please ensure that all numbers presented in the abstract are present and identical to numbers presented in the main manuscript text.

AUTHOR SUMMARY

* Under ‘What did the researchers do and find?’, we suggest briefly explaining in lay terms what the energy landscape analysis does.

INTRODUCTION

*Please remove the study results and/or conclusion from the Introduction.

*Please conclude the Introduction with a clear description of the study question or hypothesis.

METHODS AND RESULTS

* Since most of our readers will not be familiar with energy landscape analysis, we have found that the description of the results can be quite complex. When revising the manuscript, please consider explaining the results in a broadly accessible way, focusing on the most relevant findings.

* “In pn-TTC Wave 3” – was wave 3 in 2020?

* In the description of the study participants, please ensure that it’s 100% clear what data from which time point was included for the analysis in your study focusing on the energy landscape analysis as well as the MRI analysis.

* “We will release the Python code in the near future.” - Please make any custom code available, either as part of your data deposition or as a supplementary file. Also, please include the statement on code availability in the data availability statement in the online submission form.

* “Image acquisition and processing” – did the MRI analysis include data from all four waves (2013-2023) or just data from July 2019 to October 2021? Please clarify.

* Please clarify whether the pn-TTC received ethical approval prior to commencing the first wave 1 in 2013.

* Figure 3: Please define G1 and G2. Please add the definitions for Periods 1-4. Please ensure that you add sufficient detail and explanation in the figure description (as done in Figure 2). Each figure should be self-explanatory on a stand-alone basis.

* Figures: Please define all elements of box plots in the figure caption - center line, box limits and whiskers. Please indicate in the figure caption the meaning of the whiskers in Figure (e.g. 95% CI or SD).

* Figures: We think it would be useful to briefly explain the meaning of the states 111111 and 000000.

* At times, e.g. ll.550-556, we find that the results section contains too many re-iterations/explanations of methods details. Please ensure that the methods are sufficiently explained in the Methods section and in a way that’s accessible to a broad readership.

* Please consider avoiding the use of red and green in order to make your figure more accessible.

* Figure 4: Please define G1 and G2. For graphs C-H, please mention the type of mention the type of plot.

DISCUSSION

* “disturbing reports from Japan” – please avoid using emotive language.

General Editorial Requests

---

## [Decision Letter · Decision Letter 3]

17 Dec 2025

Dear Dr. Iwami,

Thank you very much for re-submitting your manuscript "Evolving landscape of psychological distress among Japanese high school students during the COVID-19 pandemic" (PMEDICINE-D-24-03134R3) for review by PLOS Medicine.

Thank you for your detailed response to the reviewers' and editor’s comments. There are a few remaining comments from the statistical reviewer and a few minor editorial issues that need to be addressed before we can accept the manuscript for publication. When submitting your revised paper, please once again include a detailed point-by-point response to the editorial comments. Please revise the paper accordingly, and submit the final revision by December 24. The remaining issues that need to be addressed are listed at the end of this email.

In revising the manuscript for further consideration here, please ensure you address the specific points made by each reviewer and the editors. In your rebuttal letter you should indicate your response to the reviewers' and editors' comments and the changes you have made in the manuscript. Please submit a clean version of the paper as the main article file. A version with changes marked must also be uploaded as a marked up manuscript file. Please also check the guidelines for revised papers at http://journals.plos.org/plosmedicine/s/revising-your-manuscript for any that apply to your paper.

A reminder that when your manuscript is accepted, an uncorrected proof of your manuscript will be published online ahead of the final version, unless you've already opted out via the online submission form. If, for any reason, you do not want an earlier version of your manuscript published online or are unsure if you have already indicated as such, please let the journal staff know immediately at plosmedicine@plos.org.

Please note that the journal will operate at reduced capacity from December 22 to January 2. If you have any questions in the meantime, please contact me (atosun@plos.org) or the journal staff on plosmedicine@plos.org.

We look forward to receiving the revised manuscript.

Sincerely,

Alexandra Tosun, PhD

Senior Editor 

PLOS Medicine

plosmedicine.org

Comments from Reviewers:

Reviewer #2: The authors have addressed all comments comprehensively, and the manuscript now represents a robust and valuable contribution to the field. I am pleased to recommend this work for publication and look forward to its positive impact. I have only a few minor suggestions that may further strengthen the manuscript:

1) Consider including a minimal predictive or cross-validation analysis, or more clearly acknowledging this limitation.

2) Summarize key LCA model indicators (e.g., fit statistics, class sizes) in the Results section to complement the supplementary figures.

3) Ensure that all interpretations remain appropriately cautious, particularly regarding causal language.

4) Provide a slightly expanded intuitive explanation of the energy landscape approach to support accessibility for non-technical readers.

Requests from Editors:

* Please note that the title should include the study design (i.e., after a colon). Editorial suggestion: Psychological distress among Japanese high school students during the COVID-19 pandemic: An energy landscape analysis

* Abstract: “The evolving energy landscape revealed that the pandemic reduced the likelihood of being in a depressive state.”- We suggest moving the sentence after ‘….healthy state increased to 18.2, 18.5, and 15.0 times that of the depressive state, respectively”.

* Please define t, p, and q the first time used (‘caudal part of the middle frontal gyrus (cMFG) (t=-2.36, p=0.019, q=0.048)).

* Please include the statement on code (“The programming code we used to calculate the energy landscape is a Python version of the MATLAB code available from [30] along with a tutorial. The Python code is available on GitHub (https://github.com/ttttmmttddiikk/Energy_Landscape_Analysis) and Zenodo (DOI: 10.5281/zenodo.17585043).”) in Data Availability statement in the online submission form.

* Regarding comment 1) from Reviewer #2: While adding a minimal predictive or cross-validation analysis would be beneficial, the editorial team would accept if you acknowledged this as a limitation.

---

## [Editor Report · Decision Letter 4]

19 Dec 2025

Dear Dr Iwami, 

On behalf of my colleagues and the Academic Editor, Alexander C. Tsai, I am pleased to inform you that we have agreed to publish your manuscript "Psychological distress among Japanese high school students during the COVID-19 pandemic: An energy landscape analysis" (PMEDICINE-D-24-03134R4) in PLOS Medicine.

Thank you for your engagement and thorough responses to the reviewers' and editors' comments throughout the editorial process.

PRESS

Sincerely, 

Alexandra Tosun, PhD 

Senior Editor 

PLOS Medicine